



# Volcanism and climate change as drivers in Holocene depositional dynamic of Laguna del Maule (Andes of central Chile - 36°S)

Matías Frugone-Álvarez[1,2,3,*], Claudio Latorre[1,2,3], Fernando Barreiro-Lostres[2,4], Santiago Giralt[5], Ana Moreno[2,4], Josué Polanco-Martínez[6,7], Antonio Maldonado[8,9], María Laura Carrevedo[2], Patricia Bernárdez[10], Ricardo Prego[11], Antonio Delgado Huertas[12], Magdalena Fuentealba[1,3], and Blas Valero-Garcés[2,4]

[1]Departamento de Ecología & Centro UC Desierto de Atacama, Pontificia Universidad Católica de Chile, Santiago, Chile.
[2]Laboratorio Internacional de Cambio Global, LINCGlobal PUC-CSIC-UFRJ.
[3]Instituto de Ecología y Biodiversiday (IEB), Santiago, Chile.
[4]Instituto Pirenaico de Ecología (IPE-CSIC), Zaragoza, Spain.
[5]Institute of Earth Sciences Jaume Almera (ICTJA-CSIC), Barcelona, Spain.
[6]Econometrics Research Group, Institute of Public Economics, University of the Basque Country, Bilbao, Spain.
[7]Basque Centre for Climate Change (BC3), Leioa, Spain.
[8]Universidad Católica del Norte, Coquimbo, Región de Coquimbo, Chile.
[9]Centro de Estudios Avanzados en Zonas Áridas. Universidad de La Serena, La Serena, Chile.
[10]Instituto de Investigaciones Marinas (IIM-CSIC), Vigo, Spain.
[11]Marine Geosciences and Territorial Planning, Edificio de Ciencias Experimentales, University of Vigo, Vigo, Spain.
[12]Instituto Andaluz de Ciencias de la Tierra (IACT-CSIC), Granada, Spain

**Correspondence:** Matías Frugone-Álvarez (matutefrugone@gmail.com)

**Abstract.**

Late Quaternary volcanic basins are active landscapes from which detailed archives of past climate, seismic and volcanic activity can be obtained. A multidisciplinary study performed on a transect of sediment cores was used to reconstruct the depositional evolution of the high-elevation Laguna del Maule (LdM) (36°S, 2180 m asl, Chilean Andes). The recovered 5 m composite sediment sequence includes two thick turbidite units (LT1 and LT2) and numerous tephra layers (23 ash and 6 lapilli). We produced an age model is based on nine new $^{14}$C AMS date, existing $^{210}$Pb and $^{137}$Cs data and the Quizapú ash horizon (CE 1932). According to this age model, early Holocene were followed by a phase of increased productivity during the mid Holocene and higher lake levels after 4.0 ka BP. Major hydroclimate transitions occurred at ca. 0.5, 4.0, 8.0 and 11 ka BP. Decreased summer insolation and winter precipitation due to a southward shift in the Southern Westerly Winds and a strengthened Pacific Subtropical High could explain early Holocene lower lake levels. Increased biological productivity during the mid-Holocene ($\sim$ 8.0 to 6.0 ka) is coeval with a warm-dry phase described for much of southern South America. Periods of higher lake productivity are synchronous to higher frequency of volcanic events. During the late Holocene, the tephra layers shows compositional changes suggesting a transition from silica-rich to silica-poor magmas at around 4.0 cal ka BP. This transition was synchronous with increased variability of sedimentary facies and geochemical proxies, indicating higher lake levels and increased moisture at LdM after 4.0 cal ka BP, most likely caused by the inception of current ENSO/PDO-like dynamics in central Chile.





## 1 Introduction

The Andes of central Chile (or South-Central Andes, SCA) are prone to drought, and future scenarios of global warming show
them becoming drier as winter droughts become more recurrent (Falvey and Garreaud, 2009; Stocker et al., 2013; Boisier et al.,
2016). Many studies have documented major productivity terrestrial ecosystems, atmospheric and oceanic circulations changes
associated with dry-wet periods during the Holocene on the west slope of the Andes in Chile (e.g., Jenny et al., 2002; Lamy
et al., 2001; Martel-Cea et al., 2016; Latorre et al., 2007, 2006; Frugone-Álvarez et al., 2017; Fletcher and Moreno, 2011;
Kim et al., 2002; Kaiser et al., 2008) but the nature, regional distribution and timing of these variations in the mediterranean
regions of central Chile are not yet fully understood. Several coupled atmosphere-ocean mechanisms, including variations in
tropical Pacific Sea Surface Temperature (SST) gradients, poleward displacement of the Southern Westerly Wind (SWW) and
changes in the intensity and position of the Southern Pacific Subtropical High (SPSH), have been hypothesized to explain this
climate variability. Furthermore, little is known about the long-term evolution of large-scale climate change in the Southern
Hemisphere owing to the lack of high-resolution paleoclimatic data sets (Villalba et al., 2009).

The SCA are also one of the most active volcanic and seismic zones on Earth, due to the subduction of the Nazca plate
underneath the South-America plate (Dewey and Lamb, 1992). Recent volcanic activity, with two major eruptive events in the
last 200 years from the Quizapú Volcano (plinian eruption with a VEI index = +5, Fontijn et al. (2014)), does show that their
is a hazard to regional in central Chile. With 60+ m of permanent deformation since last deglaciation the magmatism in LdM
has been reconstructed based on geochemistry and dating of surface volcanic formations (Hildreth et al., 2010; Andersen et al.,
2017; Singer et al., 2018). In recent decades, the LdM volcanic field has shown extremely high rates of deformation (Andersen
et al., 2012; Feigl et al., 2014; Singer et al., 2018; Wespestad et al., 2019) (Figure 1) that can be interpreted as a response to
shallow magma intrusion and thus indicative of intensified volcanic hazard.

Here, we present a seismic survey and multiproxy sediment core transect study from LdM to investigate depositional vari-
ability in a lake located in one of the most active volcanic fields in the world. Although intense volcanic and seismic activity
pose a challenge to paleoenvironmental reconstructions, they also provide an opportunity to investigate the complex interplay
of these factors in the lake evolution. Geochemical proxies are interpreted in terms of biological productivity and detrital input
variation associated with climate variability and the influence of volcanic activity in the lake basin. Pollen proxies provide in-
formation about regional and local vegetation dynamics. The LdM lacustrine sedimentary sequence contains a detailed archive
of volcanic history and exemplifies how volcanic activity can impact lake processes, not only in terms of tephra input, but also
on local geochemical and biological cycles. As the sequence spans the entire Holocene, the LdM record also provides one of
the few opportunities to test previous hypotheses regarding the coupled atmosphere-ocean mechanisms that control the climate
evolution of central Chile during the Holocene (Jenny et al., 2002; Lamy et al., 2001; Villagrán and Varela, 1990; Valero-Garcés
et al., 2005; Rutllant and Fuenzalida, 1991; von Gunten et al., 2009), especially with respect to the relative contributions of



the SWW, SPSH, volcanic activity and tropical Pacific SST gradients during known the Rapid Climate Changes (RCC) events
(Mayewski et al., 2004).

## 2   Regional setting

The LdM volcanic field is part of the Southern Volcanic Zone (SVZ, 39°- 34°S) in the SCA (36°S - 70°30'N, 2180 m asl; Fig. 1c.), which is one of the most active volcanic and seismic zones on Earth (Feigl et al., 2014). The most voluminous Quaternary eruption of the volcanic field occurred ca. 950 ka and formed part of the LdM basin as a consequence of the collapse of an 80 $Km^2$ elliptical caldera (Caldera Bobadilla). Between 336 and 38 ka small rhyolitic eruptions took place, culminating with a ring of 36 postglacial rhyodacite and rhyolite coulees and domes over the last 25 ka (Andersen et al. (2012); Feigl et al. (2014); Singer et al. (2018); Figure 2).

The basin ($\sim 300~Km^2$) is irregularly shaped (Figure 2) due to both tectonic/volcanic processes and glacial erosion (Singer et al., 2000). The maximum elevation of the watershed reaches 3940 m asl at Cerro Campanario and the mean elevation is 2450 m asl (Figure 2). The lake has $\sim 50$ m maximum water depth, and the Maule River is its only outlet (Figure 2). The lake level is mainly controlled by the influx of snowmelt through small inlets and runoff. A dam was completed at the lake outlet in 1957 and as a result, water storage increased from 600 $Hm^3$ to ca. 2000 $Hm^3$ with a 30 m lake level increase (Carrevedo et al., 2015). Despite limited available limnological data, the local climate and basin morphology suggest that LdM is a dimictic lake (Frugone-Álvarez, 2016). The temperature profile in March 2013 (austral autumn) revealed that LdM was slightly stratified, with a gradual temperature decrease from 12°C at the surface to 10.5°C near the bottom (Fig. S3b). The waters were oligotrophic, with relatively high alkalinity ($[CO_3Ca]$= 424 mg/kg$^2$), pH between 7.0 and 8.4, and low concentration of salts, with nitrate and sulfate values reaching 50 $\mu$g/l and 7000 $\mu$g/l, respectively (Frugone-Álvarez, 2016). Relatively high alkalinity could be related to elevated levels of volcanic CO2 (Singer et al., 2014). Total phosphate was very low 10 $\mu$g/l, which may potentially limit biological activity. The $\delta D_{[VSMOW]}$ and $\delta^{18}O_{[VSMOW]}$ values from small inlets and water samples at different depths in the lake suggest that lake waters are derived mainly from precipitation and snowmelt with little deviation to the local meteoric water line (Fig. S3b).

Regional climate is characterized by $\sim 1700$ mm annual rainfall and a large seasonal temperature range (Fig. S4). Winter precipitation, mostly as snow, is associated with the incursion of cold fronts that migrate with the SWW (Garreaud, 1992; Falvey and Garreaud, 2007). Summers are dry due to a strong influence of the SPSH and easterly storms are infrequent (Viale and Garreaud, 2013). On interannual to interdecadal timescales, rainfall in subtropical central Chile is sensitive to variations in the El Niño-Southern Oscillation (ENSO), the Southern Annular Mode (SAM) and the Pacific Decadal Oscillation (PDO) (Jacques-Coper and Garreaud, 2014; Garreaud et al., 2009; Montecinos and Aceituno, 2003; Pittock, 1980; Quintana and Aceituno, 2012). These climate modes have a large influence on the patterns of variability of the snowpack and streamflow in central Chile along with the occurrence of extreme events, such as protracted drought or torrential rains (Masiokas et al., 2010, 2012).



Modern vegetation surrounding LdM is dominated by sub-shrubs and cushion species corresponding to the high Andean Shrubland belt (between 2000 and 2500 m asl.; Luebert and Pliscoff (2006)). At higher elevations (> 2500 m asl), the high Andean steppe characterized by scattered and/or scarce vegetation dominated by herbs and cushion species as *Oxalis adenophylla* and *Pozoa coriacea*. Low Andean shrublands occur at elevations between 1500 and 2000 m asl and include *Chuquiraga oppositifolia*, *Discaria articulata*, *Laretia acaulis*, *Berberis empetrifolia* and *Discaria chacaye*. Deciduous forest is found below 1500 m asl. (Fig. S5).

## 3 Materials and methods

### 3.1 Seismic surveys and coring

The LdM sedimentary basin was surveyed with an Edgetech 424-SB sub-bottom multi-frequency profiler using a frequency range of 2-10 kHz. Approximately 20 km of reflection-seismic data were acquired in the northern areas of the basin, with a denser grid network close to the coring sites. The data were processed and interpreted using the EdgeTech Discover SB3200 XS software. The cores were linked to the seismic data with a depth–time conversion, assuming an acoustic velocity of 1500 ms$^{-1}$ and using the density values and the p-wave velocity measured in the cores with a GEOTEK Multi-Sensor Core Logger (MSCL). Coring campaigns were organized during the summers of 2011, 2012 and 2013. We recovered 17 short cores and over 40 m of long sediment cores at several sites using a hammer-modified UWITEC gravity corer and a UWITEC platform with a percussion piston corer (Figure 2; Table S1).

### 3.2 Core Analyses

Sediment cores were transported to the IACT-CSIC laboratory in Granada (Spain) to measure physical properties (P-wave velocity, magnetic susceptibility, electric resistivity and gamma density) with a 1-cm resolution by a Geotek Multi-Sensor Core Logger (MSCL). The cores were then imaged with a Geotek Single Track Core Imaging System (MSCL-CIS) to 50 micron pixel sizes at the IPE-CSIC labs.

Sedimentary facies were defined and characterized based on macroscopic (color, bedding features, grain size, lithology and sedimentary textures) and microscopic (smear slides) criteria formulated by Schnurrenberger et al. (2003). A composite sedimentary sequence (4.6 m long) was constructed with the short cores and long cores at site 3 LEM13-3A, LEM13-3E and LEM13-3B and also LEM13-2C at site 2 used for geochemical and pollen analyses (Tabla S1).

Elemental geochemistry included total carbon (TC), total inorganic carbon (TIC), total organic carbon (TOC=TC-TIC), total sulfur (TS) at 1 cm resolution performed with a LECO elemental analyzer and total nitrogen (TN) at 5 cm resolution in a VARIO MAX CN elemental analyzer, from the Pyrenean Institute of Ecology (IPE-CSIC). Biogenic opal (BioSi) was measured using the wet alkaline leaching procedure and dissolved silica was photometrically determined using a continuous flow analyzer AutoAnalyser Technicon II at the Instituto de Investigaciones Marinas (IIM-CSIC), Spain (Bernárdez et al., 2005). The carbon and nitrogen isotope analyses were performed on bulk organic matter from sediment samples of 20-30 mg





(dry weight) at 5 cm intervals and measured in the Laboratory of Biogeochemistry and Applied Stable Isotopes (LABASI) of the Pontificia Universidad Católica de Chile using a Thermo Delta V Advantage IRMS coupled with a Flash2000 Elemental Analyzer.

An AVAATECH X-Ray Fluorescence II core scanner with a Rh x-ray tube from the University of Barcelona was used to obtain XRF logs from the LEM11-3A, LEM11-3E and LEM11-3B cores at 4 mm of resolution. Only elements with mean values higher than more than 1000 counts per second (cps) were used in the statistical analyses. We use a Robust Principal Component Analysis for dimensionality reduction (see 1.2 Supplementary Methods; Fig S7). Thirty-one samples were measured using an Inductively Coupled Plasma-Optical Emission Spectrometry (ICP-OES) at the CEBAS-CSIC laboratory in order to calibrate

the XRF qualitative measurements (Supplementary Fig. S10). Three element ratios have been calculated to describe changes in the redox conditions (Fe/Mn), carbonate (Ca/Ti) and organic productivity (Br/Ti) (Carrevedo et al., 2015; Frugone-Álvarez et al., 2017; Naeher et al., 2013; Moreno et al., 2007).

Sixty-four pollen samples were prepared, identified and counted using reference collections available at the CEAZA, Chile (see Carrevedo et al. (2015); Faegri et al. (1989)). Data were analyzed and plotted using the R package analogue (Simpson,

2007). Pollen ratios were calculated using the formula $(a-b)/(a+b)$ where $a$ is *Amaranthaceae* or *Ephedra* and $b$ is *Poaceae* (Maher, 1963, 1972).

### 3.3 Chronology

The chronology for the LdM sequence was obtained by [210]Pb and [137]Cs dating (Carrevedo et al., 2015) and nine new AMS radiocarbon dates on wood, terrestrial macrofossils, algae macrofossil and bulk sediment (Table 1). Radiocarbon ages were de-

termined by AMS [14]C at the Poznan (Poland), the DirectAMS Laboratories (USA) and UC-Irvine (Keck Radiocarbon Facility). Waters from the hypolimnion and modern living littoral macrophytes were also sampled to assess modern [14]C reservoir effect (Table 1). Radiocarbon ages were calibrated using the Southern Hemisphere Calibration curve (SHCal13) (Hogg et al., 2013). We used the R package Bacon (Blaauw and Christen, 2011) to age-depth modelling and establish the deposition rates along the core after removal of instantaneous depositional events such as tephra layers and turbidites. Radiocarbon ages are reported as

conventional radiocarbon years BP (relative to CE 1950). The short core LEM12-3B was sampled in the field for [210]Pb/[137]Cs dating every 0.5 cm for the uppermost 20 cm and at 1 cm for the lower section (Table S3). Analyses were performed at the St Croix Watershed Research Station Laboratory, Science Museum of Minnesota. Date and sediment accumulation rates were established using the constant rate of supply (CRS) model (Appleby and Oldfield, 1978). The identification of the Quizapú ash layer (CE 1932) and the appearance of *Pinus* pollen served as further chronological markers, as *Pinus radiata* was not

forested in large-scale plantations in Chile until CE 1930 (Table S3). Dates for the LdM eruptive history based on [14]C dating of paleosols, [40]Ar/[39]Ar of lava flows and [36]Cl dates or paleoshoreline exposures were compiled from the literature (Andersen et al., 2017) to better constrain the timing of the depositional evolution of LdM.



## 4 Results and Interpretation

### 4.1 Sedimentary facies

Sediments in LdM are composed of diatom-rich muds, silts and oozes with abundant interspersed volcanic facies (cryptotephra and tephra as ash and lapillis). Presence of cm-thick macrophyte and diatom-rich silt facies characterizes deposition in littoral, shallower (<25 m water depth) cores (e.g. LEM11-3A, ∼24 m water depth) (Figures 2 and 3). In more distal, deeper areas (>25 m water depth; sites 1, 2 and 3), sediments are made up of banded to laminated organic and diatomaceous silts, and include several thick massive layers interpreted as lacustrine turbidites. LdM cores from sites 1, 2 and 3 have been correlated

using sedimentary facies, volcanic layers, physical properties (MS and density) (Figure 3). Six distal hemipelagic facies, two mass wasting depositional events and 29 volcanic layers (6 lapilli and 23 ash) have been identified in the sedimentary sequences of LdM (Figures 3 and 4).

The upper 5 units in site 1 are similar to site 3 (Figure 3), suggesting that the composite stratigraphy of site 3 is representative of the sedimentary infill in the northern areas of the basin. Core LEM13-1B (31 m water depth) reached 5.8 m beneath the lake

floor, recovering 2.5 m of lacustrine sediments and volcanic facies below laminated unit 5. Unit 6 (LT2) in the shallow, western site 1 is composed of gravels and sands with abundant silty matrix, topped by a thin, homogeneous layer. Sediments below unit 6 in site 1 have been grouped in stratigraphic unit 7 (Figure 3), including coarse clastic facies, fine lacustrine laminated sediments, a thin lapilli (L6, 25 cm) and five ash layers (T19 to T23). The base of the core in Site 1 is composed of coarse breccia with angular cm-long volcanic clasts in a volcanic matrix that likely represents brecciated volcanic facies emplaced in

the distal areas of the lake. Laminated diatomaceous facies cover this basal breccia with an intercalated thick, black, maphic-rich ash (T23). The presence of homogeneous coarse sands with clay/silt matrix and coarse gravels with unsorted sandy matrix and rounded clasts suggests fluvial/alluvial depositional processes in the lake.

### 4.1.1 Hemipelagic diatomaceous Facies

Lacustrine diatomaceous facies have been classified according to elemental composition (TOC, TS, BioSi, TIC), grain size

(textures), sedimentary structure (lamination) and biological composition (diatoms and macrophytes) in three groups: i) banded silts with low TOC (< 2 %) (D1 and D2), ii) banded organic-rich silts (>2 % TOC) (D3, D4 and D5) and iii) laminated organic-rich silts (D6) (Figures 4 and S5, and Table 2). Banded, organic-rich silts occur in the upper half of the sedimentary sequence at sites 1, 2 and 3 (Facies D1 to D5) whereas laminated facies D6 occur in the lower part.

Table 2 summarizes the main characteristics of the LdM lacustrine facies. The finer grain size of facies D1 and D2 and the

absence of littoral components (i.e macrophyte remains), with variable clastic input, particularly higher during deposition of D2. Coarser grain size and the abundance of macrophytes remains setting for facies D3 compared to D1 and D2. Facies D3, D4 and D5 are organized in dm-thick sequences and they are macrophyte-dominated (D5, D3) to diatom-dominated environments (D4) (Fig. S5). Laminated facies D6 have the highest TOC and TS values up to 5.5 and 7.0 %, respectively, and contain euhedral crystals of endogenic calcite about 10 $\mu$m in size.





Biogenic silica concentrations are between 5 and 26 % with a mean of 20.5 %. A boxplot of BioSi distribution in the sedimentary facies (Figures 4 and S5h) shows the highest values in banded facies D3, D4 and D5, and relatively lower values in laminated (D6) and core top sediments (D1 and D2). D3 and D4 facies also show higher values than macrophyte-rich D5. Well-defined peaks throughs in BioSi low BioSi values occur at the base (432-433 cm), middle (364-366 cm), and top (313-316 cm) of unit 5, some of them likely associated to volcanic facies (Figure 4).

### 4.1.2    Mass Wasting Deposits and Lacustrine Turbidites

Two thick intervals of homogeneous sediments occur in the middle (Facies LT1) and at the base (Facies LT2) of the sequence (Figure 3 and Table 3). They show relatively constant XRF intensities, $\delta^{13}$C and $\delta^{15}$N values but the largest C/N range (Figures 4, S5, S8 and S9) pointing to a mixture of terrestrial, macrophytes and phytoplankton organic matter. LT1 presents a banded basal interval, coarse grained of 30 cm thick and a thicker (up to 1 m) homogeneous upper interval (Figures 4, S5
and S8). LT2 is composed of finer black silts than LT1, while the latter are browner, coarser, with more abundant macrophyte remains and mm-size pumice clasts than the former. Although the core at site 3 did not reach the base of this unit (Figure 3), the lower part of LT2 shows a small increase in MS (Figures 4 and S5) suggestive of graded texture. The upper limits for both LT1 and LT2 are sharp. LT2 is overlain by finely laminated, diatom-rich facies and LT1 by a tephra layer (T7).

The sedimentological features (gradational textures with basal layering and homogeneous at top) and the homogeneous
chemical composition are key criteria for establishing these layers as mass wasting deposits emplaced in the deeper parts of LdM. Seismic profiles (Figure  6) show some massive deposits (MTDs, landslides) related to LT1 and focussed to the deepest parts of the basin, while the lacustrine facies have a much more draped appearance (Figures 5b, S1 and S2) supporting this interpretation.

This double structure within the turbidites (banded and massive in LT1 and high basal MS in LT2) has been described in
lacustrine turbidites in the south-central Chile (Van Daele et al., 2015; Moernaut et al., 2014, 2009), the Alps (Lauterbach et al., 2012) and the Pyrenees (Corella et al., 2014). Additionally, the higher TIC, BioSi, $\delta^{13}$C and macrophyte abundance of remains in LT1 point to higher contribution of littoral sediments (Figures 4, S5 and S11). Deposition of LT1 occurred after the complex volcanic event centered around Lapilli # 3 (T9-L3-T8; Figures 3 and 4, see below) and that could explain the higher content in pumice clasts. The association with volcanic layers at base and top of LT1 suggests that volcanic activity and related local
seismicity could have been the main trigger for sediment destabilization in the lake margins.

### 4.1.3    Volcanic facies

At Site 3, distinctive volcanic facies occur as lapilli (5 L layers) and ash (18 T layers) (Table 4). Ash layers have been classified into two main groups (Fig. S6) according to macroscopic features (grain size, color, depositional structures), microscopic observation of smear slides and compositional data from XRF analyses:
1) Dark colored (T1, T2, T4, T5, T11 and T14) with relatively high content of mafic minerals and low quartz and plagioclase (Frugone-Álvarez, 2016). Semi-quantitative XRF data show higher Sr, Ti, Fe and lower Si (and thus lower Si/Ti ratio), K, and





Rb (Figs. S6a-b, S8a). T4 has the highest Ca values. This group includes most layers in the upper part of the sequence after Lapilli 1 (T1 to T5).

2) Light colored (Grey) (T3, T6, T7, T8, T9, T10, T12, T13, T15, T16, T17 and T18) with relatively high glass content and
Si (high Si/Ti ratio), low Fe and Sr, and variable Rb (Figs. S6a-b, S8a). This group mainly includes layers in the lower part of the LdM sequence.

These ashes are laterally continuous throughout the northern and central areas of the basin where we have a transect of cores. The presence of some depositional textures as normal grading and lamination, and the absence of current-derived features are all indicative of fall-out deposits from regional and local volcanic activity (Fisher and Schmincke, 1984; Sáez et al., 2007).

In the upper part of the sequence, cm-thick, volcanic-rich, dark brown to grey, homogeneous silts with gradational upper and lower limits occur. They are composed of a mixture of volcanic (glass, silicate crystals, mafics) and organic (diatoms, macrophytes, amorphous) components. They are interpreted as cryptotephras, deposited in the distal areas of the lake by fluvial and run-off processes affecting the watershed and washing out the volcanic material after the eruptions.

Lapilli occurs as thick (from 6 to 20 cm; see Figure 3) layers with up to 1 cm long pumice clasts, low matrix content, sharp
upper and lower boundaries and massive, banded or graded (both fining and coarsening upward) textures. Texture, composition and magnetic susceptibility identify two groups: the younger deposits L1 and L2 have more heterogeneous clasts in composition and size and they have higher MS, Ca, Fe and Sr values (more than 200 10-6 SI) than the older deposits L3, L4 and L5 (MS less than 100 SI) (Figures 3, 4, S6c-d and S8). The lapilli layer L3 shows the largest variability in thickness from 2 cm at the eastern sites 3E and 3B to 15 cm at the westernmost site 3A (Figure 3), suggesting a western source for this volcanic material.
It is also the only lapilli with a basal ash layer (T9).

## 4.2 Lithological and Seismic Stratigraphy

Five seismic units have been defined based on the characteristics of seismic reflectors and correlated with lithostratigraphic units at sites 2 and 3 (Supplementary Sec. 2.2). Seismic penetration reached up to 10 m in most areas, although only in the upper part, well-defined reflectors can be traced all over the northern part of the basin (Figure 6). The total thickness of the
sedimentary infill and morphology of the substratum of LdM could not be established since the substratum/infill boundary could not be seismically imaged. The available seismic profiles show the complex basin structure of LdM, with several sub-basins with variable accumulation rates, a number of faults affecting the lower part of the sediment sequence and lateral changes in thickness and physical properties (Figures S1 and S2). Using the magnetic susceptibility and density measured in the sediment cores and considering a constant acoustic velocity of 1500 $ms^{-1}$, the main reflectors in the seismic profiles were correlated
with the lithostratigraphic sequence (Figure 6).

Unit S1 uniformly drapes across the entire northern basin, without significant modification in thickness or physical properties even in the northeastern bay (LEMB10). According to the seismic-to-core correlation of the seismic profiles LEMA3 (Figure 6), unit S1 encompasses U1 to U3 lithostratigraphic units (Figure 3). Seismic unit S2 and S4 have transparent to low-amplitude seismic facies, and these correlate with stratigraphic unit 4 (Lacustrine Turbidite LT1) and unit 6 (LT2), respectively. A deposits
without a coherent internal structure and with a positive topography occur pretty close to site 3 related to units 4 and 6





(Figure 6), which can probably be interpreted as MTDs/landslides. Some profiles show complex internal structure in marginal areas in S2 (Supplementary Sec. 2.2). Seismic unit S3 is characterized by closely spaced, well-defined although not very high-amplitude reflections, wavier than in S1. Unit S3 correlates with stratigraphic unit U5.

### 4.3 X-ray fluorescence (XRF) Core Scanner geochemistry

The first two principal components (PCs) of the XRF core scanner dataset for lacustrine facies (volcanic facies excluded) explain more than 80 % of the variance (Table 12 and Figures S7 and S8). The eigenvector associated with the higher eigenvalue (5.1) define two main groups of loadings: i) Ti (0.33), Rb (0.39), Zr (0.39), K (0.31), Ca (0.35) and Sr (0.32) and, ii) As (-0.30) and Fe (-0.23). The second largest eigenvalue (1.99) is defined mainly by Br (0.50), Mn (0.62) and Ti (-0.40). The first eigenvector show positive coefficients with higher contents of clastic-related elements, and consequently, they are interpreted as

increased clastic delivery to the lake (mostly silicates from the volcanic watershed). Negative $Z_1$-score values are associated to higher values of As and Fe, and interpreted as more dominant oxic bottom conditions during periods of lower sediment delivery to the lake (Table S2). This component could also reflect water input into the system, with positive values of $Z_1$-score during more humid periods (higher sediment delivery) and negative values during less humid periods (lower sediment delivery). The $Z_2$-score is interpreted as an index of bioproductivity with positive coefficients indicative of increased total organic productivity

(and/or preservation), and negative values during periods with lower bioproductivity/preservation (Figures S7 and S8 and Table S2).

The Fe/Mn ratio has the lowest values in laminated facies D6 (lowest Fe, highest Mn; Figures S8 and S9j-h-i) and the highest values in banded facies D1-D5 (highest Fe, lowest Mn; Figures S8 and S9j-h-i). The Ca and Ca/Ti profiles generally follow the TIC curve (Figures 4, S8, S9 and S10b), less in LT2 by the high Ti values, with higher values in unit 5 than in upper units

(Figures S5b, S9d-e). Br shows a significant second order relationship with TOC ($R^2_{boostraping}$=0.42 ± 0.16, p<0.001; n=100), although this relationship is stronger for sediment samples with TOC < 4 % (Figure S10f). Br and TOC indicators have the lowest values in banded facies D1 and D2, intermediate in D3, D4 and D5 and higher in laminated D6 (Figures S5c, S9a-b). Facies LT1 and LT2 have intermediate to high Ti, K, Rb, Zr and Si values (Figures 4, S8a and S9), all indicative of minerogenic input to the lake.

### 4.4 Carbon and nitrogen stable isotopes


The $\delta^{13}$C in bulk organic matter samples range between - 15 and - 30 ‰, although most values are between - 22 and - 28 ‰ (Figures 4 and S5e) in the range of lacustrine phytoplankton and macrophyte organic matter (Meyers, 2003, 1994). Average values are lower for banded facies (D1, D2, D3, D4 and D5) and higher for laminated facies (D6) (Figure S5e). Values are also higher for LT sediments, particularly for LT1 (> - 20 ‰); (Figure S5e). The $\delta^{13}$C vs C/N ratio boxplot shows

higher (less negative) $\delta^{13}$C values corresponding to higher C/N suggesting that the type of organic matter (macrophytes versus phytoplankton) is a major control of isotopic values in bulk OM in LdM (Figures S5e and S5g).

The $\delta^{15}$N values range between - 1 and 3 ‰ (Figures 4 and S5f), with the lowest values in LT1 and LT2, intermediate in laminated facies D6 and highest in banded facies (D1 to D5). Although no isotope data for modern flora of LdM are available,





values that we have obtained from soil, particulate organic matter (POM) in the water, macrophytes and sediments of the nearby
El Piojo lake (Figure 2) range between -1 to 3 ‰, 14 to 18 ‰, 2 to 5 ‰ and - 1 to 2.5 ‰, respectively. Recent sediments in
LdM are similar to facies D1 and D2 in cores (Figure S5) and they show TOC/TN values between 12 and 14, $\delta^{13}$C from -
14 to - 20 ‰ and $\delta^{15}$N from - 1 to 10 ‰, with $\delta^{15}$N values more positive in offshore than in littoral sediments. The close
correlation (Figure S11) with sedimentary facies and organic composition (algal versus macrophyte) suggests that depositional
environments (more littoral versus more distal) play a significant role in N dynamics in LdM

## 4.5 Pollen

Pollen spectra are primarily dominated by Poaceae and secondly by *Ephedra*, *Amaranthaceae* and *Asteraceae* (Figure S12).
Poaceae are dominant throughout, especially in unit 3 and *Amaranthaceae* become dominant after unit 3. *Ephedra* dominant
principally in the part upper of unit 5. A major change occurred at 300 cm (Figures 4, S12a), marked by a significant increase
in *Ephedra/Poaceae* and *Amaranthaceae/Poaceae* ratio, which can be interpreted as an upward shift of lower vegetation belts.

## 4.6 Age model

We used four approaches to establish the age-depth model of the LdM record and the recent $^{14}$C reservoir effect: (1) dating the
water-dissolved inorganic carbon (DIC) at the mixolimnion ($\sim$20 m), (2) dating modern submerged macrophytes (Table 1), (3)
comparing ages from the same stratigraphic interval obtained from macrophytes, terrestrial samples with no reservoir effect
(i.e. wood) and tephrochronological marker such as the top ash layer in the sequence identified as the Quizapú Volcano eruption
of (CE 1932) and (4) we dated only samples of macrophytes through of the sequence (see Table 1, Figures 5, S13). According
to this age-depth model, the LdM sequence spans the last 13 cal ka BP with a reservoir effect of 4.7 cal ka BP. The age model
is thus robust for the upper 3 units (Figure 5) including $^{14}$C dates from terrestrial macrofossils, $^{210}$Pb and $^{137}$Cs dates and
the Quizapú ash and a better constrained $^{14}$C reservoir effect. The CE 1963 $^{137}$Cs peak centered at 6.5-7 cm (Figure 5b) fits
well the $^{210}$Pb chronology, and the Quizapú ash horizon a 14-15 cm, adding considerable confidence to the age model for
these upper units. Units 1 and 2 span the last 0.65 cal ka BP with an accumulation rate of $\sim$ 10 a $cm^{-1}$ and $\sim$ 20 a $cm^{-1}$,
respectively. Analyses of pollen in the LdM sequence show the first appearance of *Pinus* pollen at 30 cm (ca. 1750-1700 CE)
similarly to other pollen records in central Chile (Villa-Martínez et al., 2003; Frugone-Álvarez et al., 2017) and the sharp
increase of *Pinus sp.* pollen at 10 cm depth (Table 3A) coincides with the promulgations of the forest law in 1931 that had a
large impact in deforestation (e.g, in El Maule Province reached 143.450 ha in 1943 CE). unit 3 (195-43 cm) spans between 4.0
and 0.65 cal ka BP with a median rate of 23 a cm-1. unit 4 (LT1, 300-195 cm in site 3) started at 4.0 cal ka BP. Laminated unit
5 (440-300 cm) is characterized by a low accumulation rate ( 70 a cm-1) and includes most of the Holocene (13.0 to 4.0 cal
ka BP), whereas unit 6 (Lacustrine Turbidite 2) was deposited prior to 13 cal ka BP. The age of the lower part of the sequence
(units 5 and 6), however, is not as well constrained.



## 5  Discussion

### 5.1  Establishing a chronological controls to the lacustrine sequence LdM

Obtaining a robust absolute chronology for the LdM sequence is hampered by the lack of plant macrofossils and a large [14]C reservoir effect (Figure 5 and Table 1). Large reservoir effects are common in Andean volcanic lakes, likely due degassed magmatic $CO_2$ (Holdaway et al., 2018; Sulerzhitzky, 1971; Valero-Garcés et al., 1999). Volcanic $CO_2$ rising from sublacustrine springs is assimilated by submerged and aquatic vegetation resulting in dates that are apparently too old (Christenson et al., 2015). Although the top ash layer has not been geochemically fingerprinted, its estimated age according to [210]Pb techniques coincides with the Quizapú Volcano eruption (CE 1932) and it can be used as a chrono-stratigraphic marker in the lake sequence (Carrevedo et al., 2015). The DIC-based reservoir effect was similar to that found in living macrophytes sampled at the dam (age ca. 2.4 cal ka BP), but considerably lower than the estimate based on the comparison of samples from the same stratigraphic level: a macrophyte sample (LEM11-3A, 13 cm, ca. 4.8 cal ka BP) and a wood sample (LEM13-3D, 14 cm, ca. 0.85 cal ka BP). This variable range of the reservoir effect underlines the complexity of the carbon cycle in volcanic lakes with likely temporal and spatial variability and also suggests that biological effects on the littoral versus distal environments may be significant. A detailed study of [14]C ages variability in modern sediment, aquatic vegetation, organic producers is needed to understand these differences. With the available data and as most dated samples in our sequence were macrophyte remains, we have established a reservoir effect of 4.7 cal ka BP for the sequence resulting from the difference in age among macrophytes and wood samples from the same level (Carrevedo et al., 2015). We are aware of the simplification of these methods to estimate the reservoir effect for the whole sediment sequence, as the complexity of the carbon cycle in volcanic lakes is likely to cause spatial and temporal variability in these values. However, until a more detailed tephrochronology is developed for LdM, this methodology offers the best approach to estimate the age of the sequence. We have considered the uncertainties of the age model in the paleoenvironmental and paleoclimate implications of the LdM record.

### 5.2  Depositional dynamic of LdM

According to the genetic classification of volcanic lakes by Christenson et al. (2015) Laguna del Maule corresponds to a Caldera Lake: the lake was developed in a polygenetic volcanic system (G1), the relationship between volcanic processes and lake formation is strong (R1), the duration of the lake water fill after the eruption is long (T0) and the lake fills a large part of the caldera (L1). The main stages in the depositional evolution of LdM have been characterized based on the sedimentological and geochemical features of the composite sediment sequence for Site 3. Seven main depositional phases/events have been identified during the evolution of LdM, corresponding to the emplacement of turbidites LT2 (Phase I) and LT1 (Phase V), deposition in a shallower lake with intense volcanism, and high organic productivity (Phase II to IV) and a deeper, fluctuating and less productive lake (Phase VI and VII).



### 5.2.1 Phase I: Emplacement of turbidite LT2

The $^{14}$C – based chronology of the LdM sedimentary sequence favors a pre-Holocene (ca. 13.3 cal ka BP) for the emplacement for LT2. Andersen et al. (2017) dated the rle lava at the north end of the lake (Figure 2) as $19 \pm 0.7$ ka. This volcanic eruption dammed the lake, raising its level by 200 meters and forming a prominent shoreline around the entire basin (Hildreth et al., 2010). The dating of these paleoshoreline outcrops with cosmogenic 36Cl provided minimum age ranges of $9.4 \pm 0.4$ ka, $8.8 \pm 0.6$ ka, $7.5 \pm 0.3$ ka, $6.6 \pm 0.6$ ka and $4.2 \pm 0.2$ ka ago, suggesting that the lake drained catastrophically in the early Holocene

or later. Using these cosmogenic $^{36}$Cl dating, Singer et al. (2018) conclude that the age of this event of the paleoshoreline is ca. 9.4 ka. This is a much younger age (3500 years) compared to the age provided by our $^{14}$C model (ca. 13 ka BP).

Sedimentary facies at the base of the recovered sequence do not show evidence for deep hemipelagic deposition. According to our $^{14}$C age model, brecciated volcanic facies in Site 1 (unit 7) would be pre-Holocene and could correspond to the Volcanic Phase 1 defined by Singer (2014). Alternating hemipelagic and coarse sands with rounded clasts suggest strong flu-

vial/alluvial transport during a lake phase occurring before the catastrophic pre-early Holocene drainage of the lake (Singer, 2014; Singer et al., 2018). However, as we see no important depositional changes (i.e. from deeper to shallower facies), hiatus or erosional surfaces in the lacustrine sequence, the sediments recovered in LdM sequence could only have been deposited after this catastrophic event.

Emplacement of lacustrine turbidite LT2 could be related to three main triggering processes; (1) intense seismic and volcanic

activity (Moernaut et al., 2019) in the basin during the late glacial ($^{14}$C age model), (2) intraplate (intraslab or crustal) earth-quakes (Van Daele et al., 2019; Wils et al., 2018) related to a postglacial/Holocene uplift (Singer et al., 2018) in combination with tectonic processes of the Troncoso Fault or (3) to the event drainage of the lake if this emplacement would have been related to the catastrophic drainage of the basin, it should have occurred later, according to the $^{36}$Cl-dating of the highest lake shoreline ca. 9.4 cal ka BP (Singer et al., 2018). The discrepancies between the two age models may only be resolved with

additional dating methods (e.g. with individual lipid terrestrial compounds or estimating a dynamic reservoir effect through record) and the development of a tephrochronology for the LdM sequence. Indeed, the occurrence of tephra layers in the LdM sequence provides a unique opportunity for fingerprinting and linking them to the known $^{40}$Ar/$^{39}$Ar and $^{14}$C dated eruptions in the basin (Andersen et al., 2017; Singer, 2014; Singer et al., 2018).

### 5.2.2 Phase II: A shallow, low productivity lake

Unit 5 represents deposition in the LdM between two major catastrophic events: LT2 (late glacial to early Holocene, according to our $^{14}$C age model) and LT1 (mid Holocene, 4 ka BP). After deposition of LT2 turbidite, fine lacustrine deposition resumed in the northern LdM basin with diatomaceous facies D6 (Sites 1, 2 and 3; Figure 3). At Site 3, sediments overlying LT2 are only about 30 cm thick (unit 5, 411-438 cm depth) and composed of facies D6a (less diatomaceous and less organic-rich) and D6b (more organic-rich and with some endogenic calcite). This alternation reflects transition from diatom-dominated productivity

(D6a) to macrophyte-dominated (D6b), likely caused by lake level fluctuations (Figure 7). A major volcanic event deposited L5 and several tephra (T18 and T17) (Table 5). Higher $Z_1$-score values suggest a relatively higher supply of volcanic detritus



from the catchment to the lake that will continue during the next phase (Figure 7). The sharp drop in Fe/Mn values indicates a rapid change to more oxic conditions at the lake bottom or a large change in water geochemistry as a result of the volcanic activity. Several studies have shown that the Fe/Mn ratio in the sediment is influenced by the redox conditions at the sediment-
water interface at the time of deposition (Frugone-Álvarez et al., 2017; Moreno et al., 2007; Naeher et al., 2013; Wersin et al., 1991). More dominant reducing conditions at the water sediment interphase would enhance dissolution of Mn (II) but would not greatly affect Fe (II) leading to higher Fe/Mn sediment values (Davison, 1993; Tipping et al., 1981). Differences in the input of Mn and Fe to the sediments weathering intensity and volcanic activity. Higher sediment input from the watershed and more oxic conditions at the bottom of the lake are coherent with lower levels and increased mixing regimes in the lake.
Most indicators (TOC, TIC, C/N, $Z_2$-score, Br/Ti) suggest a change in the amount and composition of phytoplankton and terrestrial organic matter (Meyers (2003), Figure 7). At this time, BioSi reached the lowest values of the whole sequence suggesting lower diatom productivity (Figure 7). The $\delta^{13}C$ in bulk organic matter values are in the same range, although slightly more negative than other Andean lakes (e.g., Pueyo et al., 2011; Díaz et al., 2016; Contreras et al., 2018) where sources of heavy carbon attributed to magmatic $CO_2$ inputs seem to play a significant role in C cycling (Valero-Garcés et al.,
1999). The $\delta^{15}N$ values depend on contributions of algae, macrophytes and land plants, and the nitrogen source ($NO_3^-, N^2$) (Botrel et al., 2014). $\delta^{15}N$ is also used as an indicator of primary productivity (Meyers, 2003) with less positive values linked to an decrease in phytoplankton productivity during this phase Meyers and Teranes (2002). Pollen samples indicate sparse vegetation and relatively high *Ephedra/Poaceae* ratio would suggest relatively humid conditions facilitating an upward shift of lower vegetation belts. Therefore, the pollen point to a shift to increased aridity at the base of unit 5 and a return to more humid
conditions in upper units 3 to 1 (Figures 7 and S12).

### 5.2.3 Phase III: A carbonate-producing, moderate productivity lake

The dominance of facies D6c indicates a large depositional change in the lake basin. Sediments with higher content of macrophyte remains, higher TIC and Ca/Ti values, and the presence of small crystals of endogenic calcite and sulfur-rich minerals (TS, $\sim 2$ to 7 %, Figure 7) marks the onset of this phase. The $Z_1$-score values remained high indicative of maintained high de-
trital input. Although with some peaks, the relatively lower Fe/Mn ratio suggest dominant oxic conditions at the sediment/water interface (Figure 7). Sedimentological and geochemical evidence points to an alkaline and oxic environment, which contributed to carbonate formation and sulfur oxidation (Figure 7). During this phase, the C/N values show peaks with the highest values in the sequence ($\sim 15$); moderate BioSi indicate relatively lower phytoplankton productivity (Figure 7). Fluctuations in TOC, C/N, $\delta^{15}N$, BioSi, Br/Ti and $Z_2$-score point to periods of improved environmental conditions for macrophyte growth (more
available littoral settings) reflect a climate shift towards warmer conditions. The close relationship with sedimentary facies and organic composition (phytoplankton versus macrophytes) suggests that the depositional environment (more littoral versus more distal) plays a significant role in C and N dynamics at LdM (Fig. S11). The decrease in TOC and the less abundant calcite occurrences indicate a less productive environment towards the end of this phase, simultaneous the decrease in *Ephedra/Poaceae* values indicative of increased aridity (Figure 7).





Carbonate deposition in another Andean volcanic lake — Lago Chungará, 18°S, 4500 m asl (Figure 1) — has been exten-
sively studied (Pueyo et al., 2011; Sáez et al., 2007; Moreno et al., 2007) and provides a comparison for this site. LdM is an
alkaline volcanic lake (Pecoraino et al., 2015), with pH > 8, relatively high $HCO_3^-$ and $CO_3^{2-}$ and low $Ca^{2+}$ concentrations.
Several factors control carbonate formation in volcanic Andean lakes, (i) variations in salinity due to evaporation; (ii) the input
of calcium due to weathering of new volcanic material; (iii) $CO_2$ photosynthetic depletions related to seasonal phytoplankton
blooms; (iv) the development of littoral settings more favorable to charophyte growth; and (v) changes in C cycling due to
volcanic activity. The source of Ca in LdM is likely the weathering of the andesitic/basaltic and rhyolitic rocks either from
andesitic/basaltic lithologies of the older Cola de Zorro Formation or late Glacial-early Holocene rhyolitic eruptions (Hildreth
et al., 2010). Increased precipitation of calcite and higher presence of sulfate minerals during this phase could be related to the
synergistic effects of increased volcanic activity — demonstrated by the higher number of tephra layers intercalated in the se-
quence — and favorable environmental conditions conducive towards increased weathering and Ca input to the lake (Figures 7
and 8). The increased volcanic activity during phase III could correspond to the early Holocene volcanic phase defined by
Andersen et al. (2017). Higher temperatures during the early Holocene could have also been a significant factor for promoting
chemical weathering of the surrounding volcanic rocks, while in Chungará Lake the role of salinity due to the aridity could be
a more decisive factor.

### 5.2.4   Phase IV: A high carbonate, more productive and macrophyte-dominated productivity lake

Organic-rich, carbonate-bearing facies D6a dominated deposition in the central areas of LdM during this phase. Bioproductivity
proxies (TOC, Br/Ti and $Z_2$-score) show a trend with elevated values at the beginning of this phase and after a decreasing up-
core trend (Figure 7). Moderate BioSi values and decreasing C/N reaching the lowest values in the sequence suggest a period
of phytoplankton dominance as organic producers during the beginning of this phase. Dominant oxic conditions continued and
together with a decreased minerogenic (lower $Z_1$-score) and lower *Ephedra*/*Poaceae* values also indicate water mixing, lower
runoff and drier conditions, respectively (Figure 7). This phase ended with higher detrital input (higher $Z_1$-score) and the onset
of an increasing trend Fe/Mn, coherent with an increase in water depth and establishment of more distal environments in the
coring site. The $^{14}$C based age model suggests this phase would correspond to the mid Holocene (8-4 cal ka BP). Geochemical
proxies underscore the millennium prior to deposition of LT1 (ca. 5-4 cal ka BP) as one with the lowest bioproductivity and
the highest clastic input during the Holocene. Higher clastic input could be related to the increased volcanic activity at the end
of unit 5.

### 5.2.5   Phase V: Emplacement of turbidite LT1 (4.0 cal ka BP.)

A complex sequence of events at LdM occurred about 4.0 cal ka BP, starting with intense and prolonged volcanic activity and
deposition of a thick, multi-story volcanic unit including T9-L3-T8, followed by a thick turbidite unit (LT1) and capped by
another tephra unit (T7) (Figures 3 and 4). This volcanic event is unique in the sequence, as basal tephra T9 shows fine-grained,
coarsening upward texture and convoluted lamination that could indicate pyroclastic flow processes or transported to the lake
by runoff (Fontijn et al., 2016). This 4.0 ka BP volcanic event could be linked to activity from the SE volcanic centers in





the LdM basin during Phase 2, which started with the development of the Barrancas Sequence (6.4 ka) at the SE of the lake (Andersen et al., 2017). The thick turbidite LT1 was emplaced in the northern, deeper areas of the basin after deposition of ash

fall (T8) and capped by T7 tephra. Depositional mechanisms would have been similar to early Holocene turbidite LT2.

### 5.2.6   Phases VI: A deeper lake during the late Holocene (4.0 – 0.7 cal ka BP)

Phase VI started after the emplacement of LT1 with a completely new set of facies (banded D3, D4 and D5) characterized by a sharp decrease in $Z_1$-score values reaching the lowest values in the sequence and an increase in productivity (TOC, Br/Ti, BioSi and $Z_2$-score). Lower organic content in diatom-rich facies D4 compared to macrophyte-rich facies D3 and D5 suggests that

the littoral setting is the most productive in LdM, as shown in many other mountain lakes (Michelutti et al., 2015; Quayle et al., 2002; Smol and Douglas, 2007). Fe/Mn values also showed a sharp increased and maintained higher values until the top of the sequence (Figure 3). Lower clastic input, more anoxic conditions, higher phytoplankton productivity, higher *Ephedra/Poaceae* values, lower $\delta^{13}$C values (increase of $C_3$ plants) and an increased in facies variability at ca. 4.0 cal ka BP are coherent with more distal/deeper conditions at the coring site, higher/fluctuating lake levels and increased winter precipitation. Modification

in basin morphology and accommodation space after the 4.0 cal ka BP event would have also contributed to this depositional change. Sediment delivery ($Z_1$-score) increased starting at 1.5 cal ka BP, and exhibits a centennial-scale pattern (Figure 7). Higher frequency of TOC, Br/Ti, $Z_2$-score and BioSi values suggest that bioproductivity variability increased since 4.0 cal ka BP, reaching similar or even higher values than during some mid-Holocene intervals. This increasing variability could be driven by a ENSO-like forcing (Moy et al., 2002). Most proxies (TOC, C/N, BioSi, and $Z_2$-score) identify two main productivity

transitions at $\sim$ 2.0 cal ka BP and 0.7 cal ka BP (Figure 7). The period 4.0-2.0 cal ka BP is characterized by intermediate TOC values and high diatom productivity (BioSi) with some TIC peaks (periods of increased alkalinity); conversely from $\sim$ 2.0 cal ka BP to 0.7 cal ka BP, TOC was higher, but BioSi, Ca/Ti, C/N, Br/Ti and $Z_2$-score values decreased (Fig. 7). Higher Fe/Mn ratio variability occurred during the first period compared to the second (Figure 7). Sediment delivery ($Z_1$-score) increased starting at 1.5 cal ka BP, and exhibits a centennial-scale pattern (Figure 7).

### 455  5.2.7   Phase VII: Lower productivity and more anoxic conditions (last 7 centuries)

The depositional evolution of LdM during the last 0.7 cal ka BP has been described in detail in Carrevedo et al. (2015). The onset of this phase is marked by a decreased in bioproductivity and an increased in Fe/Mn values. After deposition of coarser, more littoral facies D3 during unit 2 ($\sim$ 0.62 cal ka BP – late $19^{th}$ century) finer, less organic-rich facies D1 and D2 were deposited during the last century (unit 1). Bioproductivity indicators (TOC, BioSi, Br/Ti and $Z_2$-score) show two century-scale

peaks from CE 1650 to 1850 and *Pinus* and *Rumex* pollen at the top of the sequence reflect recent human activity (Carrevedo et al., 2015). Low values occur from CE 1400 to 1650 as well as during most of the late $19^{th}$ and $20^{th}$ centuries. The damming of the lake in the mid $20^{th}$ century increased average lake level, favoring finer sedimentation and more anoxic conditions in the bottom of lake. An increase in lake level in the last few decades is also suggested by sedimentological and geochemical proxies.



### 5.3 Paleoclimate implications

The LdM sequence provides new insights on central Chilean paleoclimate history, particularly about the nature of the early Holocene, the timing of the mid Holocene transition at a regional scale and the nature of the larger climate variability during the late Holocene. In the following sections we will take into account the chronology uncertainties of the LdM age model, when discussing the different events in LdM, and the paleoclimate inferences will be discussed at a inter-millennial time scale.

#### 5.3.1 A relatively arid early Holocene

After the emplacement of LT2 (Phase I) and likely after the catastrophic drainage of the LdM basin (Singer et al., 2018), the lake was characterized by the dominance of shallower environments with relatively low productivity (Phase II). Soon afterwards, a large depositional change towards a more carbonate producing and organic-rich matter accumulating lacustrine system occurred with the expansion of environments adequate for macrophyte growth (littoral areas) and more dominant oxic lake bottom with well-mixed waters. During this phase, bioproductivity indicators display oscillations at millennial–scale embedded within a long-term decreasing trend (Figures 7 and 8d,e). These cycles suggest relatively lower lake levels with more macrophyte littoral productivity alternating with higher lake levels and increased planktonic productivity (Figure 7). These changes are synchronous with the decrease in *Ephedra/Poaceae* ratio that has been interpreted here as the onset of the Holocene, regionally characterized by a transition from a dry/cold climate during the late glacial to a dry/warm climate in central Chile during the early Holocene (Kim et al., 2002). A sharp increase in organic productivity and more abundant macrophytes (C/N values $\sim 17$ and $\delta^{15}$N values between 0.5 and 1 ‰) are indicative of lower water lake levels in LdM during the early Holocene that could have been caused by a decrease in snow accumulation during winter. High clastic input (higher $Z_1$-score; Figure 7) during this phase could be explained by the dominance of shallower, littoral settings in the lake. These lower lake levels, and relatively more arid conditions in the LdM record during the early Holocene, are in good agreement with other regional record from continent (Tagua Tagua, at about 150 km from LdM, Valero-Garcés et al. (2005); Jara and Moreno (2014) Figures 1 and 8) and the ocean (Lamy et al., 2010, 1999; Muratli et al., 2010a, b). Arid conditions during the early to mid-Holocene are also shown in pollen records at Quintero and Quereo (33°S) (Villagrán and Varela, 1990) and in Laguna Aculeo (34°S) (Jenny et al., 2002) from > 9.5 to 5.7 cal ka BP (Figures 1 and 8). Drier conditions during the early to mid-Holocene have also been documented at 40° − 43°S (Abarzúa et al., 2004; Moreno, 2004; Moreno and León, 2003). Sedimentary records from oceanic, fjord and lake sites from southern, central and northern Chile indicate widespread warming at southern hemisphere mid-latitudes between 12.5 and 8.5 cal ka BP (Lamy and Kaiser, 2009; Fletcher and Moreno, 2011, 2012; Kaiser et al., 2008; Kim et al., 2002; Lamy et al., 2010).

The abrupt shift to warmer conditions at the onset of the Holocene (Figures 7 and 8) could have been caused by synergetic interactions between the southward displacement of SWW and a strengthening of ENSO associated with a weakened in the Atlantic Meridional Overturning Circulation (AMOC) that would leads to a southward shift of the ITCZ (Haug et al., 2001; Kienast et al., 2006; Mosblech et al., 2012; Rein et al., 2005; Schneider et al., 2014). During the early Holocene, the SWW would have been in a more summer-like condition compared to a more winter-like pattern during the late Holocene (Lamy



et al., 2010). A positioning of the SPSH at the latitude of LdM (∼ 36°S) with an initial weakening in early Holocene and a subsequent strengthening towards the late Holocene would have caused SWW-summer conditions between ∼ 12.0 and 9.0 cal

ka BP, and decreased moisture in the Andes of central Chile (Figures 7 and 8).

### 5.3.2   A complex mid Holocene transition (8.0-4.0 cal ka BP)

Phase III likely spans the mid Holocene (8-4 cal ka BP) and represents the period with the highest organic productivity in the lake during the Holocene. Taking into account all proxies analyzed, productivity started to decrease at ca. 6.5 cal ka BP — and it became dominated by phytoplankton and *Ephedra* pollen — reaching minimum values around 4.5-4.0 cal ka BP

(Figures 7 and 8). This shift towards more phytoplanktonic productivity could reflect an increase in water depth between ca. 6 and 4 cal ka BP as littoral, macrophyte-rich environments were reduced in surface area as shown in other lakes (Nõges, 2009). Hydrological changes beginning at ca. 8-6.5 cal ka BP are also known from other regional records (Figure 8). Reconstructed precipitation from sedimentological, palynological and diatom data from Laguna Aculeo (∼ 34°S, 300 m asl) suggest more arid (150-300 mm a$^{-1}$) and warmer conditions prior to 7.5 cal ka BP, a progressive increase up to 5.7 ka with the development

of a fresher water lake (up to 450 mm a$^{-1}$), and a marked lake level rise after 3.2 cal ka BP (up to 550 mm a$^{-1}$) (Jenny et al., 2002). Geochemical and pollen records from the north-central Chilean coastal areas show an open landscape between 8.7 and 5.7 cal ka BP as a reflection of drier conditions, followed by an increase in arboreal pollen during increasingly wetter conditions between 5.7 and 3.0 cal ka BP (Frugone-Álvarez et al., 2017; Maldonado and Villagrán, 2006; Villagrán and Varela, 1990). Low lake levels and high littoral bioproductivity at LdM between ca. 8-6 cal ka BP (Figure 8) also are coincident with

decreased winter precipitation in the Andes during the mid-Holocene (Fornace et al., 2014; Moreno et al., 2007; Valero-Garcés et al., 1999, 1996). In addition, a neoglacial advance in the Mendoza Andes valley located at ∼ 35°S suggest cold conditions between 5.7 and 4.0 cal ka BP (Espizua, 2005), which is coeval with the changes that took place at LdM during this time interval (Figure 8).

Further north, the Atacama and the tropical Andes records shown a complex humidity spatio-temporal structure during the

early-to-mid Holocene and with clear differences between high altitude environments (Giralt et al., 2008; Grosjean and Núñez, 1994; Grosjean et al., 2003; Moreno et al., 2007; Sáez et al., 2007; Schittek et al., 2015; Valero-Garcés et al., 1999, 1996) and the mid altitude areas (Latorre et al., 2003; Betancourt et al., 2000; Quade et al., 2008). Intense and rapid moisture fluctuations during the mid-Holocene associated with modifications in precipitation seasonality have been proposed as an explanation for these discrepancies (Grosjean et al., 2003). South of LdM, at 46°S, increasing wind and precipitation (Daele et al., 2016)

occurred between ca. 10 and 4-5 cal ka BP; in southern Patagonia (53°S) the main decrease in precipitation also occurred between 11 and 8 ka but this decrease continued until ∼ 5 ka (Lamy et al., 2010). The lag in the climatic response from south to north is mainly a result of the expanding/displacement SWW belt toward the north.

Instrumental rainfall records from central Chile (27 - 36°S) indicate a tendency towards warm/wet (cold/dry) climate during positive (negative) phases of ENSO/PDO, respectively, that have a strong influence in the snow pack accumulation over the

subtropical and mid latitude Andes (Garreaud and Falvey, 2009; Masiokas et al., 2006; Montecinos and Aceituno, 2003). Variation in strengthening of the circulation associated with ENSO/PDO-like teleconnection (e.g., cold fronts that are controlled



atmospheres by the SWW and SPSH) could explain great part of the variability in bioproductivity in LdM: during periods of strengthened ENSO/PDO-like conditions, higher snow winter accumulation could produce higher runoff, increased lake levels, decreased littoral (macrophyte) productivity and increased planktonic productivity. On the contrary, periods of weakening

ENSO/PDO-like conditions ╍or more frequent La Niña-like conditions (for example, 4.5-4.0 cal ka BP) would have favored increased organic productivity associated to lower water level and less intense winters (Masiokas et al., 2006; Nõges, 2009).

### 5.3.3   A more humid Late Holocene (last 4 ka)

Increased bioproductivity, decreased clastic input and the onset of dominant anoxic conditions (higher Fe/Mn) in LdM at ca. 4.0 cal ka BP (Figure 7) point to higher lake levels and increased phytoplanktonic productivity. The Fe/Mn ratio shows a change

in oxidation state in the sediment/water interface at ∼ 4.5 cal ka BP to more anoxic conditions that continued throughout the late Holocene (Figure 7). Although geomorphic changes in the lake basin due to volcanic/seismic activity might have played a significant role, the coherent presence of high lake water levels in other regional records favors also a climate influence, with higher winter snowfall compared to early and mid-Holocene. All bioproductivity proxies show a distinctive transition at 2.0 cal ka BP with a decreasing productivity and a relative increase in clastic input (Figure 7)

Overall, moisture availability in central Chile during the late Holocene seems to have increased, paralleling summer insolation and seasonality which reached their maximum in the in the late Holocene (Wanner et al., 2008). The Aculeo record shows a progressive increase in effective moisture after 5.7 cal kyr BP and the establishment around 3.2 cal ka BP of modern humid conditions (Jenny et al., 2002; Villa-Martínez et al., 2003). Numerous intercalated clastic layers after 3.2 cal ka BP reflect floods during rainy winters associated to a more frequent/intense El Niño-like activity (Jenny et al., 2002). Furthermore,

paleoceanographic proxy data from a marine core off central Chile show a significant increase in palaeoproductivity and decreasing SSTs at 33°S and 41°S (Kim et al., 2002; Lamy et al., 1999) also interpreted as increased ENSO activity beginning at ca. 5.0 cal kyr BP (Rodbell et al., 1999). Higher lake levels and increased moisture availability during the Late Holocene have also been documented in the Altiplano (Laguna Miscanti, 23°S, Valero-Garcés et al. (1999, 1996); Grosjean et al. (2001) and in northwest Patagonia (Lago Pichilafquén, ∼ 40°S, Jara and Moreno (2014)). The LdM record supports increasing moisture

during the late Holocene from tropical to mid latitudes in South America. Both greater fluctuations water levels and stronger seasonality in temperature and precipitation would have prolonged seasonal cycles, and modified the hydrological dynamics of the basin in LdM. The invoked climate mechanisms include a strengthening of ENSO/PDO-like activity and a northward shift of the ITCZ (Bird et al., 2011; Carré et al., 2014; Haug et al., 2001; Moy et al., 2002; Rein et al., 2005; Stansell et al., 2013; Zhang et al., 2014) (Figure 8).

The LdM productivity record shows two main changes during the last millennium: i) around CE 1300, when a drop in organic bioproductivity and the onset of more frequent anoxic conditions occurs and ii) around the late $19^{th}$ century. The first period matches with a changes in the southern Pacific leading to cooler and wetter climate with increased storminess due to ENSO intensification, which largely affected the Polynesian societies (Rull et al., 2015). The second, and more significant, fall in productivity is concomitant to the rapid increase in *Aulacoseira* and the decrease in planktonic *Discostella stelligera* and

marks the end of the LIA (Carrevedo et al., 2015). Pollen data indicate relatively colder and moister conditions during the LIA





(Carrevedo et al., 2015). The Medieval Climate Anomaly is characterized by a higher bioproductivity in comparison at the Little Ice Age (Figures 7 and 8).

## 5.4 Holocene volcanic history based on LdM tephra deposits

The LdM age model allows to compared the volcanic activity recorded in the LdM sequence (Figure 5d) with the volcanic evo-
lution described by Andersen et al. (2017) based on surface deposits (Figure 5e), but assignment of the LdM volcanic events found in our sequence (Figure 3) to dated volcanic events (Singer, 2014; Singer et al., 2018) is not straightforward. Detailed mapping and dating of surface volcanic formations (Figure 2) have led to the identification of two main recent phases of silicic volcanism in LdM: Early Post Glacial (between 26-19 ka BP) and Holocene (Andersen et al., 2017; Singer, 2014; Singer et al., 2018). Occurrence of tephras T23 to T19 is coherent with regional volcanic activity prior to the Holocene (Figure 3). The
presence of coarser tephras (Lapilli L6, Site 1) suggests activity from closer volcanic centers. Although a genetic relationship could not be demonstrated, the relationship with LT2 suggests that emplacement of L6 occurred at the same time as major volcanic/seismic events in LdM basin (Figure 3). Intense shaking of slope sediments and instability caused by hydrological changes or sediment load related to volcanic activity could have triggered mass wasting processes resulting in the LT2 emplacement in the basin (Figure 3). This is consistent with the lower $\delta^{13}$C and C/N ratio values which are indicative of greater
content of $C_3$ land plant in the sediment (Figure S11). Central Chile is also well known for its megathrust earthquakes and many Andean lakes contain turbidites triggered by earthquake shaking (Van Daele et al., 2015). Megaturbidites as LT1 and LT2 could be triggered by large megathrust earthquakes occurring after intense volcanic activity that produced the deposit of a thick, coarse-grained tephra in the lake lowering the slope-failure threshold (Wiemer et al., 2015).

Volcanic activity in the LdM volcanic field during the early Holocene was intense (Figure 3 and Table 5). After the large
volcanic event responsible for L5, four tephra layers deposited (T17 to T14). More silica-rich composition of tephra (T18 till T9) and lower MS values of L4 and L3 suggest a dominant rhyolitic chemistry composition during this phase and it is also coherent with the end of Latest Pleistocene to Holocene volcanic phase (Hildreth et al., 2010; Singer et al., 2000; Singer, 2014; Andersen et al., 2017). Some tephra from early Holocene (T14, T15 and T16 see below) also have a distinctive composition with low XRF values of Fe and Sr (Figure S6). Interestingly, this was a period with high organic content in the sediments (more
macrophyte remains in facies D6c), highest TS and common presence of endogenic calcite, all suggesting some synergetic interactions between volcanic activity and carbonate formation in the lake (Figure 4). The second half of this volcanic phase was relatively less active, and ended with the deposition of L4 (Figure 5). Although the dating of this part of the sequence is still uncertain, the onset of calcite formation in the lake could be synchronous to the beginning of the Holocene and so this phase could encompass the early Holocene (up to 8 cal ka BP).
Lapilli L2 and L1 and ash T6 to T1 layers deposited during the last 4 cal ka BP have petrographic and chemical characteristics pointing to silica-poor composition possibly associated to rhyodacitic magma (Figure S6). Eruptive volcanism in LdM during the last 4 ka issued from eruptive centers around Cari Launa (rsl: 3.3 ± 1.2 ka; rcl: < 3.3 ka), Divisoria (rcd: 2.1 ± 1.3 ka), Laguna Sin Puerto (rdsp: < 3.5 ka), rhyodacite of the Northwest Coulee (rdcn; 3.5 ± 2.3 ka), rhyodacite of Laguna Sin Puerto (rdsp; < 3.5 ka) and Nieblas (rln: late Holocene) areas (Figures 2 and 5). Eruptions during Holocene Phase are



dominantly rhyolitic eruptions, although andesitic, rhyodacitic and mixed eruptions also occurred (Figures 2 and 5). Although, late Holocene rhyodacite eruptions of Laguna Sin Puerto and Northwest Coulee could be linked to deposition of L2 and L1 lapilli.

## 6  Conclusions

The LdM record provides a high-resolution reconstruction of depositional evolution of a volcanic lake in the South-Central
Andes during the Holocene based on sedimentological, geochemical and biological indicators. The LdM sediment sequence includes offshore lacustrine, volcanic and massive wasting deposits. Six lithostraphic units have been defined in the northern area of the basin and correlated with five seismic units. The sequence spans the Holocene, after the catastrophic drainage of the lake basin likely due to upstream erosion of the Maule River. Two mass wasting deposits are likely associated to volcanic event and/or seismic activity. Volcanic facies occur as lapilli (6 layers) and ash (23 layers). Their compositional features suggest
a late Holocene transition towards more silica-rich magma compositions. In spite of the chronologic uncertainties, the LdM record indicates lower lake levels during the early Holocene with millennial scale bioproductivity changes coherent with lower summer insolation and increased aridity. Higher bioproductivity occurred during the mid Holocene (from ca. 8.0 to 6.0 cal ka BP), synchronous with the phase of aridity described for the tropical and temperate latitudes of South America. During the late Holocene, the LdM record indicates relatively higher lake levels, consistent with increased moisture after 4.0-3.0 cal
kyr BP, caused by the inception of the current ENSO/PDO-like dynamics in central Chile. The Medieval Climate Anomaly is characterized by increased bioproductivity whereas the Little Ice Age shows a two-phase structure with cold/wet intervals between CE 1300–1450 and CE 1600–1850 interrupted by a warmer climate between CE 1450-1600. The LdM record also suggests that millennial-scale Holocene climate and water availability in central Chile was largely ruled by variations in the summer insolation. Complex interrelations between solar irradiance and dynamics changes in regional patterns of internal
climate variability such as the ENSO/PDO-like, SWW and the SPA, however, seem to exert a major control at centennial to decadal scales.

*Data availability.* The data from proxies and facies are available from the authors upon request (matutefrugone@gmail.com).

*Author contributions.* The first author named is lead and corresponding author. We describe contributions to the paper using the taxonomy provided in the title. Writing – Original Draft: Frugone-Álvarez, Matías; Valero-Garcés, Blas; Giralt, Santiago; Moreno, Ana and
Latorre, Claudio. Writing – Review & Editing: all authors. Conceptualization: Frugone-Álvarez, Matías; Valero-Garcés, Blas; Giralt, Santiago; Moreno, Ana; Josué Polanco-Martínez and Latorre, Claudio. Investigation: all authors. Field: Frugone-Álvarez, Matías; Valero-Garcés, Blas; Giralt, Santiago; Barreiro-Lostres, Fernando; Carrevedo, María Laura. Methodology: 1) Seismic surveys and coring: Frugone-Álvarez, Matías; Valero-Garcés, Blas; Giralt, Santiago; Barreiro-Lostres, Fernando and Carrevedo, María Laura. 2) Core Analyses: Frugone-Álvarez, Matías; Valero-Garcés, Blas; Giralt, Santiago and Moreno, Ana. Sedimentology: Frugone-Álvarez, Matías and Valero-Garcés. Formal Anal-



ysis: 1) Figures and tables desing: Frugone-Álvarez, Matías 2) Pollen Analysis: Maldonado, Antonio. 3) Diatom analysis: Carrevedo, María Laura. 4) Mineralogy: Giralt, Santiago. 5) AVAATECH X-Ray Fluorescence and Geochemistry: Frugone-Álvarez, Matías; Valero-Garcés, Blas; Giralt, Santiago and Moreno, Ana 6) BioSi analysis: Bernárdez, Patricia and Prego, Ricardo. 7) Carbon and nitrogen stable isotopes: Frugone-Álvarez, Matías; Fuentealba, Magdalena and Latorre, Claudio. 9) Oxygen and Deuterium stable isotopes: Frugone-Álvarez, Matías and Delgado Huertas, Antonio. Project Administration: Frugone-Álvarez, Matías and Valero-Garcés, Blas; Funding Acquisition: Frugone-Álvarez, Matías; Valero-Garcés, Blas; Moreno, Ana and Latorre, Claudio.

*Competing interests.* The authors declare that they have no conflict of interest.

*Acknowledgements.* This work was supported by the HOLOCHILL project (CGL2012-32501); the IEB (grants PFB23); grants FONDECYT 1150763; FONDECYT 1140837; FONDECYT 3180368; CONICYT PCI Project PII20150081 and ICM NC120066. A. Maldonado and C. Latorre also thank ongoing support from the Program PCI CONICYT, Proyecto Anillo SOC1405. J. Polanco-Martínez was funded by a Basque Government post-doctoral fellowship. We thank Raquel Lopez, Elena. Royo, C. Alcaino, Carlota Escutia, Ariadna Salabarnada, H. Orellana, Feña, P. Tarrats and A.L Herrera for help with sample analysis and logistical support in the field.



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



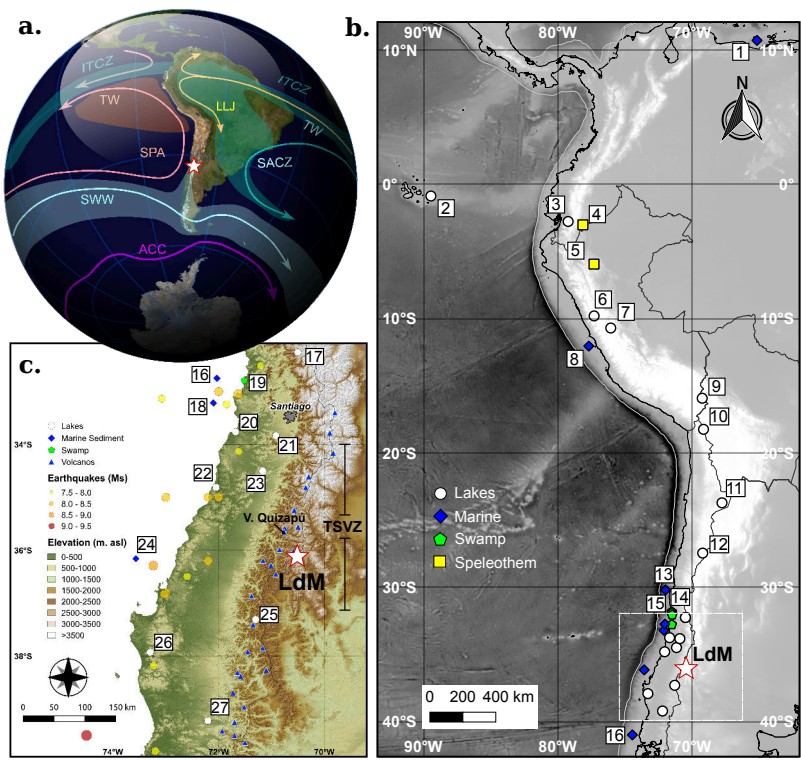

**Figure 1.** Regional context of LdM in the Central-Southern Andes of central Chile. (a) Schematic map of the principal low-level atmospheric flow over South America. SWW: Southern Westerly Wind, SPA: Southeast Pacific Anticyclone, TW: Trade Winds, ITCZ: Intertropical Convergence Zone, LLJ: Subtropical Low-Level Jet Stream, SACZ: South Atlantic Convergence Zone and ACC: Antarctic Circumpolar Circulation. Light green (tropical rainfall), orange (stratocumulus and cold SST) and light blue (mid latitude precipitation) areas represents the influence of atmospheric precipitation associated with each forcing, modified from Garreaud et al., 2009. The map (a) and (b) shows the main locations of the paleoclimate records discussed in the text with a digital elevation model (STRM30, Becker et al. 2009 and Smith et al. 1997). White circles are lacustrine records; 2: El Junco lake (Zhang et al. 2014), 3: L. Pallcacocha (Moy et al., 2002), 6: L. Queshquecocha (Stansell et al., 2013), 7: L. Pumacocha (Bird et al., 2011), 9: L. Titicaca (Fornace et al., 2014), 10: L. Chungará (Moreno et al. 2007), 11: L. Miscanti (Valero-Garcés et al., 1996;1999), 12: L. Negro Francisco (Grosjean et al., 1997; Valero-Garcés et al., 1999), 17: L. Chepical (de Jong et al. 2013), 20: L. Matanzas (Villa-Martínez, 2002) 21: L. Aculeo (Jenny et al. 2002; 2003), 22: L. Vichuquén (Frugone-Álvarez et al., 2017), 23: L. Tagua Tagua (Valero-Garcés et al., 2005), 25: L. Laja (Urrutia et al., 2010), 26: Lanalhue and Lleu Lleu lake (Stefer et al. 2010), 27: Pichilafquén lake (Jara and Moreno 2014). Green pentagons: coastal peat swamp records; 14: Ñague and Quereo (Maldonado and Villagrán, 2002; Villagrán and Varela, 1990), 15: Palo Colorado (Maldonado and Villagrán, 2006), 19: Quintero (Villa-Martínez and Villagrán, 1997). Blue diamonds: marine records; 1: Cariaco Basin ODP1002 core (Haug et al., 2001), 8: SO147-106KL (Rein et al., 2005). 13: GEOB7139-2 (De Pol-Holz et al., 2007), 16: ODP 1233 and GeoB3313-1 cores (Muratli et al., 2010; 2010b; Lamy et al., 2002), 18: GEOB3302-1 and GIK17748-2 cores (Kim et al., 2002), 24: ODP 1234 and ODP 1235 cores (Muratli et al., 2010a; 2010b). Yellow squares: speleothem records; 4: Santiago cave (Mosblech et al., 2012), 5: Tigre Perdido cave (van Breukelen et al., 2008). The software used to build the map was QGIS 2.10.





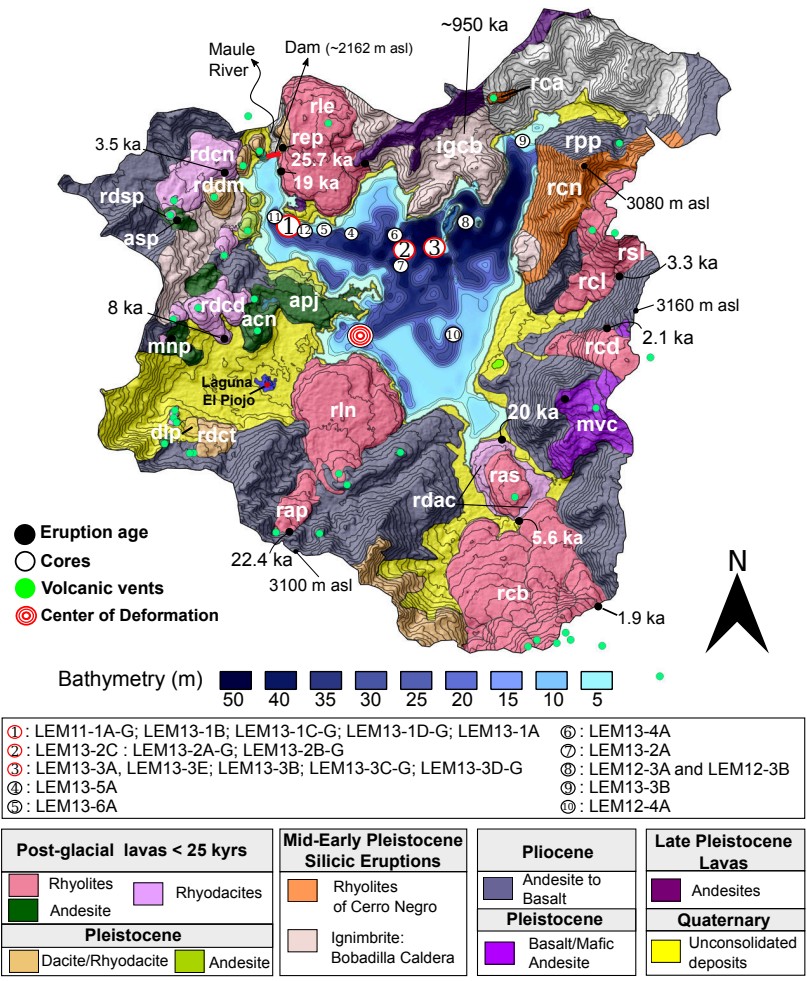

**Figure 2.** Bathymetric and geological map of LdM volcanic complex (based on Hildreth et al. (2010); Andersen et al. (2017); Singer et al. (2018)). White, green, grey and red circles indicate coring sites, volcanic vents and sample locations for $^{40}$Ar/$^{39}$Ar, $^{36}$Cl dating (Andersen et al., 2017) and El Piojo Lake, respectively. The cores used to determine the reservoir effect correction are derived from site 3 (LEM13-3D-G: $^{14}$C age for the wood) and site 12 (LEM11-3A-G: $^{14}$C age for macrophytes). Grey circles showing Eruptions preglacial (>25 ka) composed principally by the Ignimbrite of Cajones de Bobadilla (igcb; ∼ 950 ka), Rhyolite of Cerro Negro (rcn; ∼ 447 ka), Rhyodacite of Domo del Maule (rddm; ∼ 114 ka), Basalt of El Candado (bec; ∼ 63 ka) and the Andesite of Arroyo Los Mellicos (aam; ∼ 26 ka). Eruptions post-glacial (<25 ka) including the Rhyolite East of Presa Laguna del Maule (rep; 25.7 ka), the Rhyolite of Loma de Los Espejos (rle; ∼ 19 ka), Rhyolite of Cari Launa (rcl;<3.3 ka), Rhyolite south of Laguna Cari Launa (rsl;3.3 ± 1.2 ka), Rhyolite of Arroyo de Sepúlveda (ras; 20-19 ka), Rhyolite of Cerro Barrancas (rcb: multiple flows; 11.4–1.9 ka), Rhyolite of Colada Divisoria (rcd;2.1 ± 1.3 ka), Rhyolite of Colada Las Nieblas (rln; late Holocene), Rhyodacite of Arroyo de la Calle (rdac;∼20 ± 1.2 ka), Rhyodacite of Colada Dendriforme (rdcd; 8 ± 0.8 ka), Rhyodacite of the Northwest Coulee (rdcn; 3.5 ± 2.3 ka), Rhyodacite of Laguna Sin Puerto (rdsp;<3.5 ka), Rhyodacite west of Presa Laguna del Maule (rdop), Andesite of Laguna Sin Puerto (asp; <3.5 ka) and the Younger Andesite of West Peninsula (apj; 21 ± 3.4 ka). The age uncertainties are 2σ.





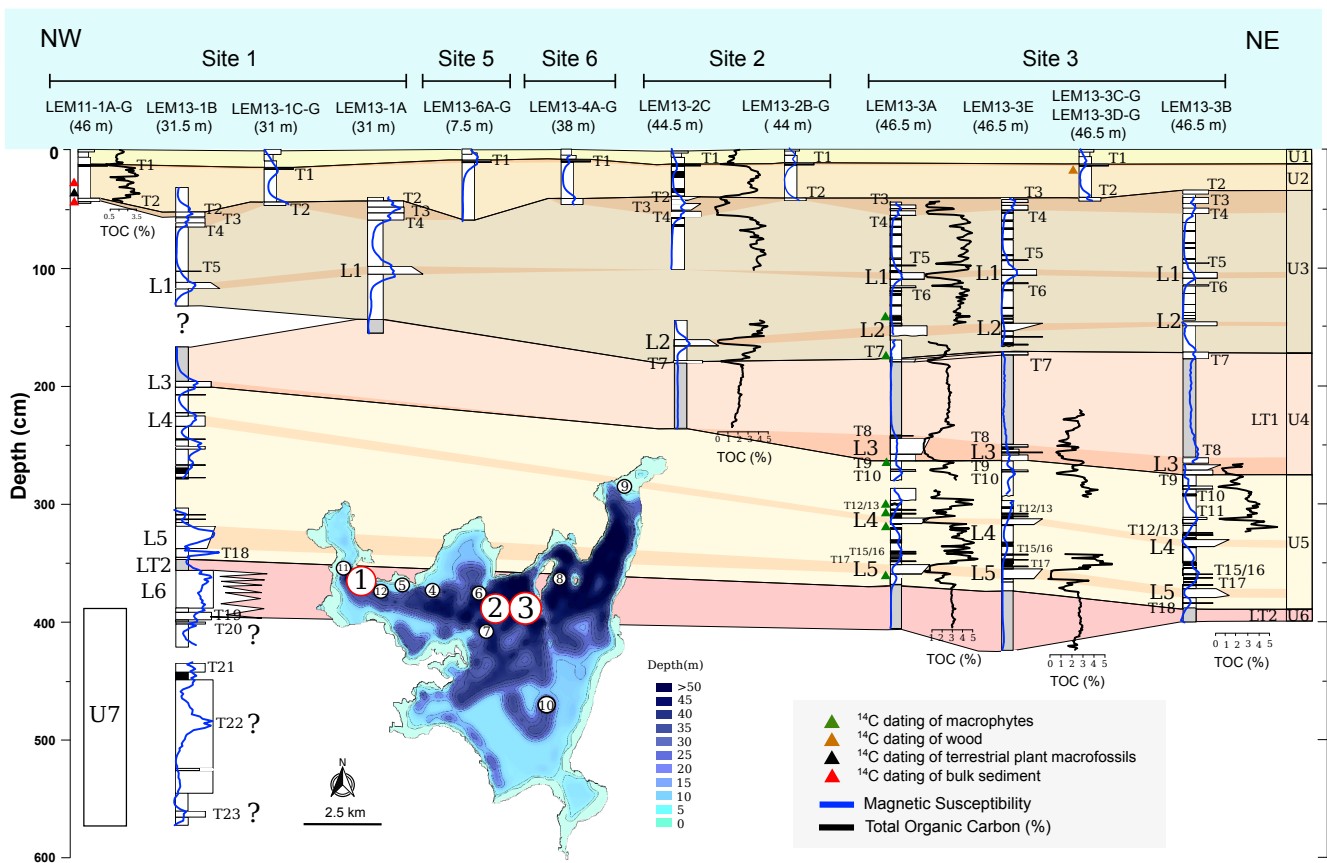

**Figure 3.** West-East LdM core correlation in the northern areas based on lithostratigraphic and sedimentological criteria. White circles indicate coring site. Magnetic susceptibility (MS [x10$^{-8}$ ($m^3/kg$)]), total carbon organic (TOC, %) and the occurrence of tephra layers. The stratigraphic units are marked in different colors; unit 1: yellow; unit 2: light brown; unit 3: brown; unit 4 (LT1): light red; unit 5: light yellow; unit 6 (LT2): red.



**Figure 4.** Sedimentary facies and sedimentological units in LdM sequence for site 3. Six lithostratigraphic units (U) and three main facies groups: Lacustrine Facies (D1 to D6); Lacustrine Turbidites (LT1 and LT2); Volcanic facies [lapillis (L1 to L5) and tephras (T1 to T18)]. The lacustrine facies have been classified according to elemental composition Magnetic Susceptibility (MS [x10$^{-8}$ ($m^3/kg$)]), percentage of the Total Organic Carbon (TOC), Total Sulfur (TS), Total Inorganic Carbon (TIC), Biogenic silica (BioSi); atomic C/N values; $\delta^{15}$N and $\delta^{13}$C values in ‰ [standardized with N2-Air and Vienna Pee Dee Belemnite (VPDB), respectively], and XRF ratios as proxies for redox conditions (Fe/Mn) and organic (Br/Ti,) productivity.





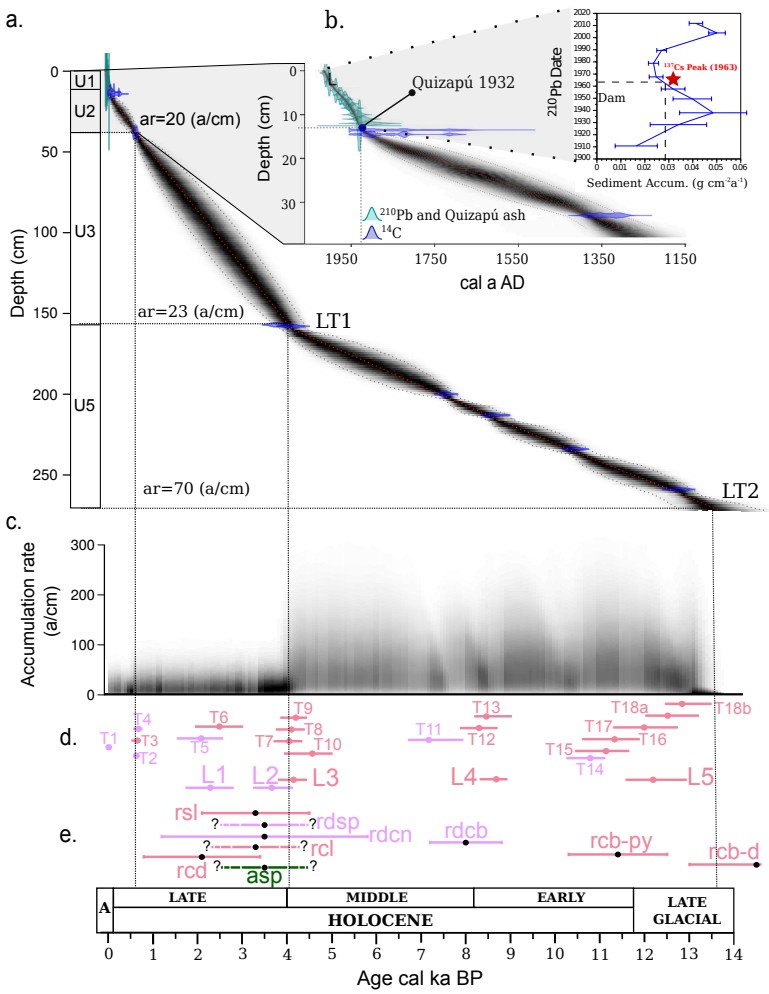

**Figure 5.** Bayesian chronological model for LdM sequence based on combined analyses of $^{210}$Pb, $^{137}$Cs and nine AMS $^{14}$C dates. Bayesian age model of LdM showing the calibrated $^{14}$C dates and the age-depth model (Blaauw and Christen, 2011). a) Chronological model for the last 14.0 cal ka BP. b) Detail of the CRS (constant rate of $^{210}$Pb supply) model, the 1963 depth determined from $^{137}$Cs peak (red star at 6.5-7 cm) and the Quizapú ash horizon (∼ 14-15 cm). c) The accumulation rates (in years per cm) as estimated by the MCMC iterations (Blaauw and Christen, 2011) with a median of 10, 20, 23 and 70 a/cm for units 1, 2, 3 and 5, respectively. d) Schematic representation of the volcanic facies ages estimated from chronological model for LdM. The ash and lapilli layers are color-coded according to macroscopic and microscopic features, and compositional data from XRF analysis; pink: ash with higher Si and Rb and lower Fe and Sr, mauve: ash with higher Sr, Ti, Fe and lower Si and Rb; mauve: lapilli layers with MS > 200 SI, higher Ca, Sr and lower Fe and K; pink: lapilli with MS < 200 SI and higher K and lower Sr, Ca and Fe. e) Distribution of the Laguna del Maule eruptive units ages reported by (Andersen et al., 2017); pink (Rhyolite), mauve (Rhyodacite) and green (Andesite) colors represents post-glacial eruptions units described on Hildreth et al. (2010); Andersen et al. (2017); Singer et al. (2018) (see Fig. 1). Note in d) and e) the greatest number of volcanic events during the late-glacial/early Holocene and middle/late Holocene transitions.





a. Seismic profiles

c. Seismic Stratigraphy LEMA 3

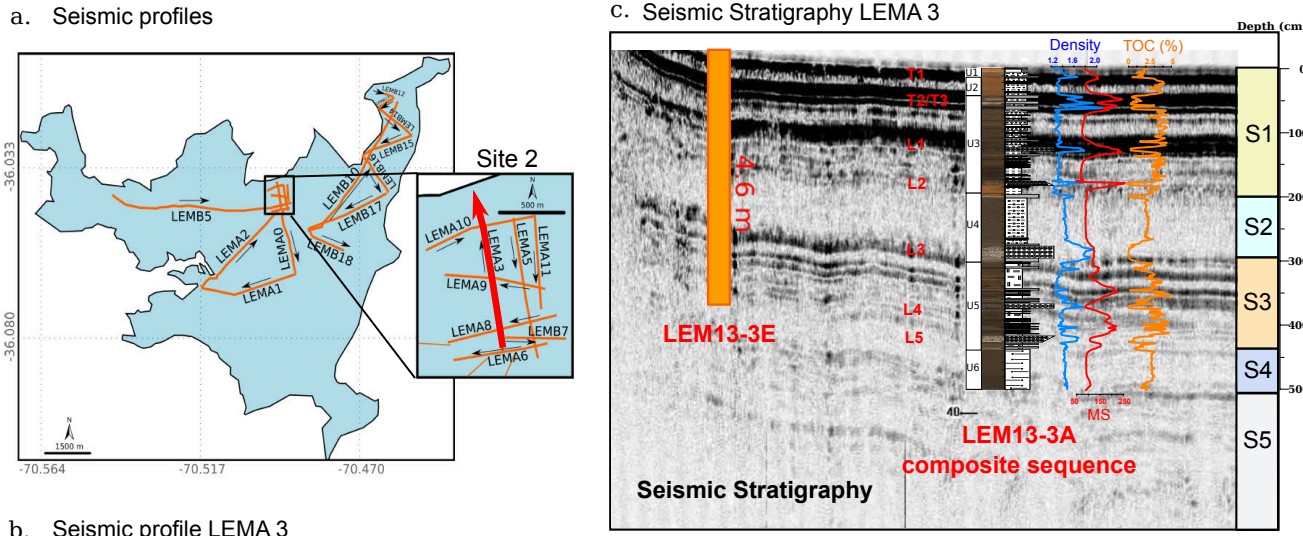

b. Seismic profile LEMA 3

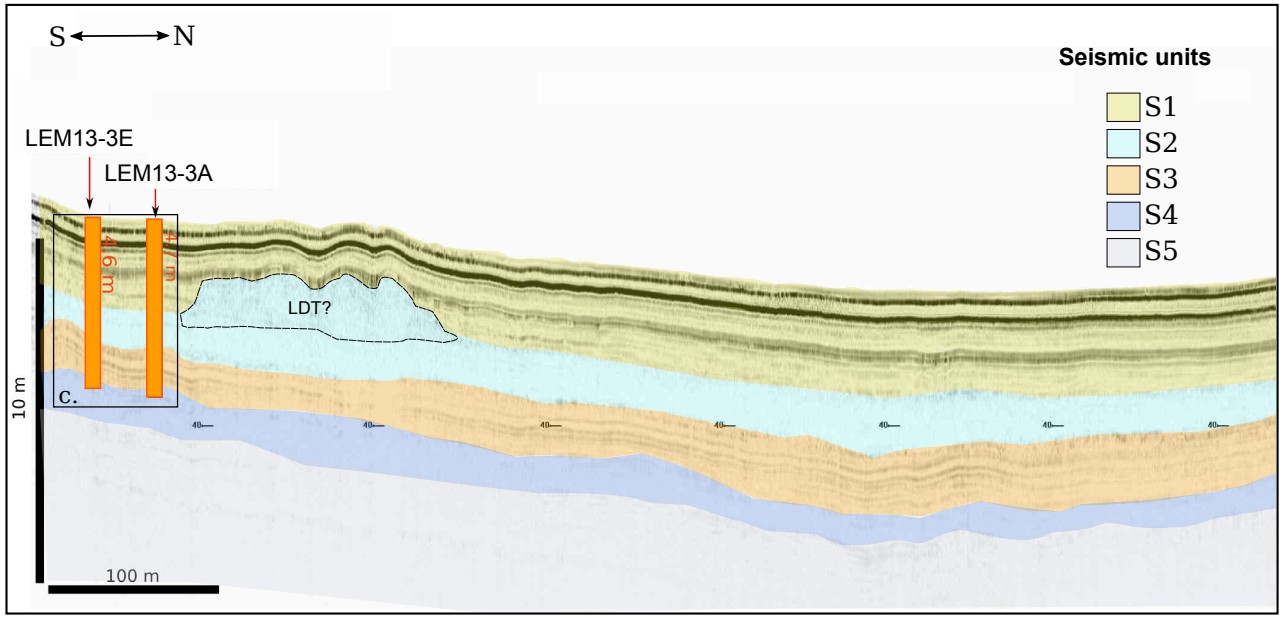

**Figure 6.** Seismic-to-core correlation and seismic to lithostratigraphic units comparison. (a) LdM map showing the location of main seismic lines with line numbers. (b) S-N 3.5 kHz seismic profile crossing LEMA 3 (red line on Fig. 6a). We recognize five different seismic units (S1 to S5), based on the characteristics of seismic reflectors and correlated with lithostratigraphic units in northern areas of LdM. Rectangles mark detailed views shown in Fig. 5c. (c) Seismic-to-core correlation of sites 2 along the cross N–S reflection seismic profile LEMA 3. The seismic units correspond to the retrieved sequence including banded to laminated lake sediments with intercalated mass wasting deposit (transparent to homogeneous seismic facies) and volcanic layers (main reflectors) and coarse lacustrine/alluvial facies at the base (see supplementary data).



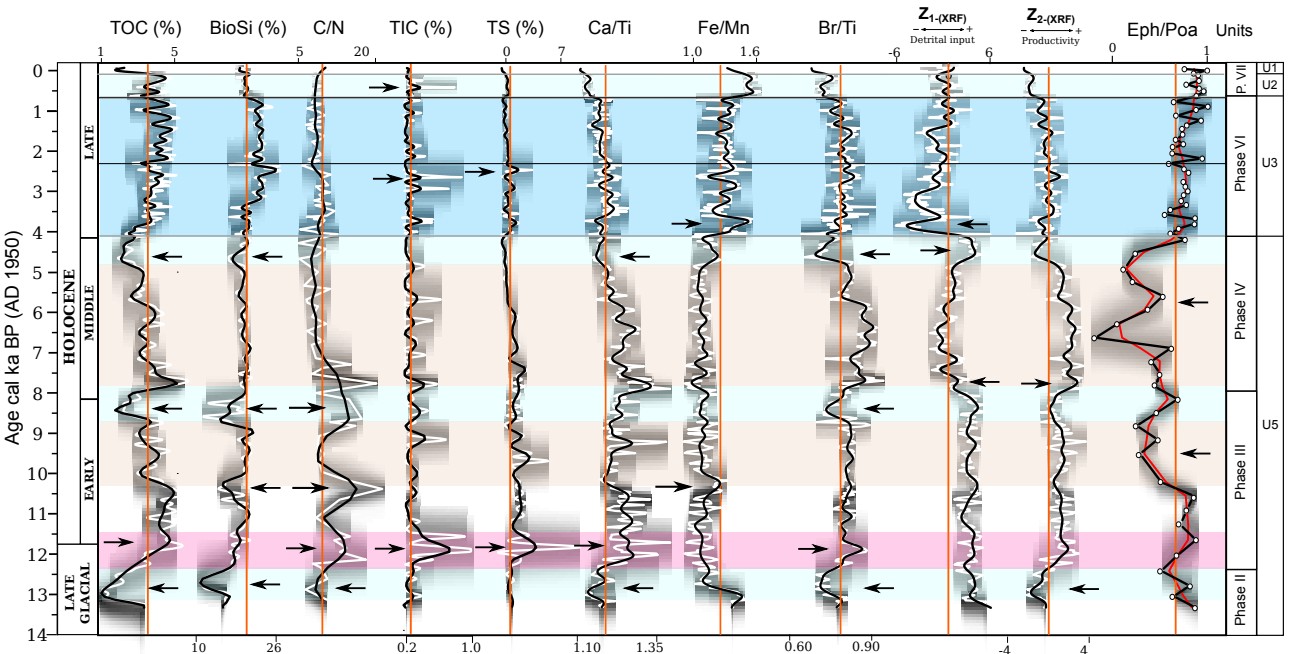

**Figure 7.** Holocene palaeoenvironmental evolution in the LdM sequence. Comparison of organic productivity, paleoredox, sediment delivery indicators ($Z_1$-score) and *Ephedra/Poaceae* (Eph/Poa) ratio plotted according to the posterior age-depth model (grey). Total Organic Carbon (TOC, %), Opal (BioSi, %), Total Inorganic Carbon (TIC, %), Total Sulfur (TS, %), and the XRF ratios in logarithm of carbonate (Ca/Ti) productivity, anoxic conditions (Fe/Mn), organic (Br/Ti and $Z_2$-score) productivity and sediment delivery indicators ($Z_1$-score). Darker greys indicate more likely calendar ages; white curve shows single 'best' model based on the weighted median age for each proxies; solid black curve shows the fit for each time series using a smoothing polynomial splines; orange lines are the average of each times series. The blue, light blue, red, light brown boxes indicate the timing of cold/humid, dry/cold, dry/warm, dry conditions recorded in the LdM, respectively.



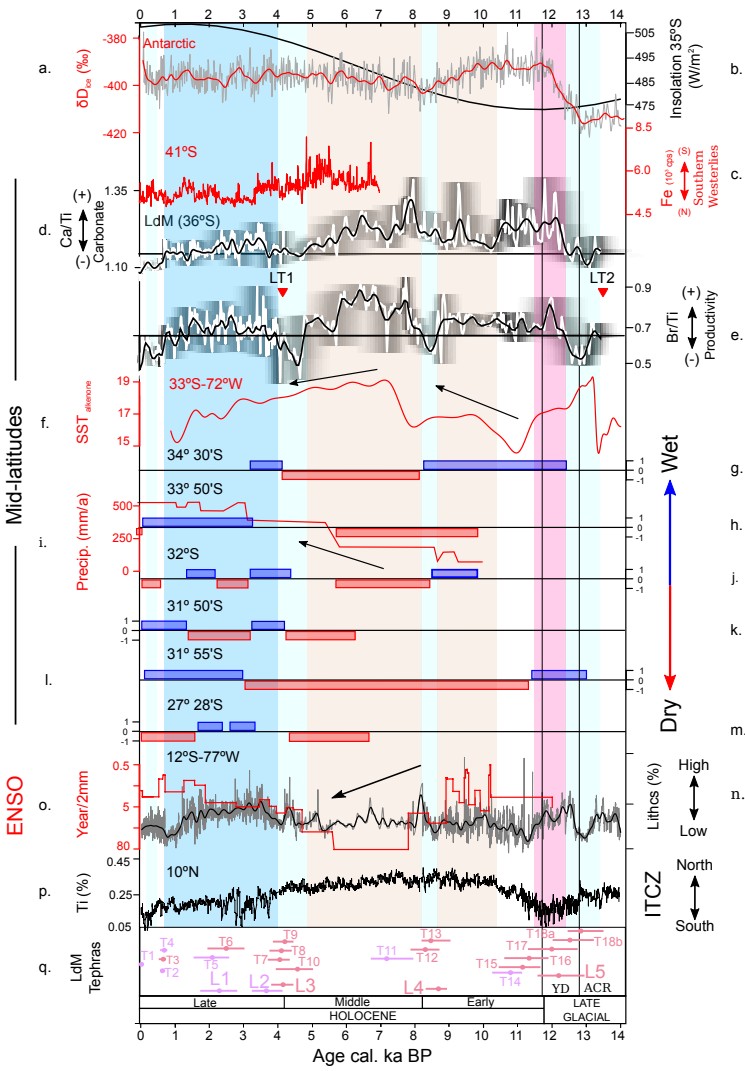

**Figure 8.** Comparison of selected proxies in LdM sequence with representative local, regional and global paleoclimate records. (a) $\delta$D Dome C ice core record (Jouzel et al., 2007). (b) Southern Hemisphere summer (DJF) insolation to 35°S (Berger and Loutre, 1991). (c) Iron contents of core GeoB 3313-1 (Lamy et al., 2002). (d) and (e) carbonate (Ca/Ti) and total organic productivity (Br/Ti) of the LdM sequence. (f) sea-surface temperatures ($SST_{alkenone}$) from the continental slope off mid-latitude Chile (Kim et al., 2002) at ~33°S. (g) to (n) categorization of the moist conditions interpreted by Holocene paleoclimate records in central Chile, where: 1; wet, 0; similar to present and -1; dry. (g) Tagua Tagua Lake (Valero-Garcés et al., 1999). (h-i) Aculeo Lake (Jenny et al. 2002; 2003). (j) Palo Colorado (Maldonado and Villagrán, 2006). (k) Ñague Swamp Forest (Maldonado and Villagrán, 2002). (l) Quereo Swamp Forest (Villagrán and Varela, 1990; Villa-Martínez and Villagrán, 1997). (m) Negro Francisco Lake (Grosjean et al., 1997). (n) and (o) Holocene marine record ENSO-sensitive from Peruvian shelf (Rein et al., 2005). (p) Bulk Ti content of Cariaco Basin sediment (Haug et al.,2001) (see Fig 1). ACR: Antarctic Cold Reversal; YD: Younger Dryas. The blue, light blue, red, light brown boxes indicate the timing of cold/humid, dry/cold, dry/warm, dry conditions recorded in the LdM, respectively.



**Table 1.** Radiocarbon dating of LdM sequence. We used only the macrophytes samples for the construction of age-depth model with a corrections of 4700 years. The table show the median age with reservoir effect corrected from the Bayesian age-depth model and uncertainties (lower and upper error). (∗) Indicates the AMS radiocarbon dates without $^{14}$C reservoir effect. (∗∗) Dating excluded from the age model. (∗∗∗) Radiocarbon age of dissolved inorganic carbon (DIC) from hypolimnion to 20 m depth and modern living littoral macrophytes used to assess modern $^{14}$C reservoir effect.

| Lab Code | ID Section | Depth(m) | $^{14}$C a BP | $1\sigma$ | Median$_{W/R}$ (cal a BP) | lower a | upper a | Material |
|---|---|---|---|---|---|---|---|---|
| Poz-59915 | LEM11-3D | 0.13 | 85 (∗) | 25 | 48 | 86 | 23 | Wood |
| Poz-57545 | LEM13-3A | 0.14 | 4,820 | 60 | 36 | 60 | 22 | Macrophytes |
| UCIAMS133686 | LEM11-1A | 0.30 | 4,760 (∗∗) | 15 | – | – | – | Bulk sediment |
| UCIAMS133687 | LEM11-1A | 0.38 | 680 (∗) | 35 | 536 | 633 | 400 | Plant macros |
| D-AMS001135 | LEM11-1A | 0.47 | 4,367 (∗∗) | 25 | – | – | – | Bulk sediment |
| Poz-59917 | LEM13-3A-2U | 1.57 | 8,230 | 50 | 4,040 | 4,395 | 3,741 | Macrophytes |
| Poz-59918 | LEM13-3A-2U | 1.58 | 8,500 | 50 | 4,066 | 4,410 | 3,790 | Macrophytes |
| Poz-59919 | LEM13-3A-3U | 2.00 | 11,440 | 60 | 7,466 | 7,800 | 6,890 | Macrophytes |
| Poz-59921 | LEM13-3A-3U | 2.13 | 12,550 | 70 | 8,517 | 8,940 | 8,157 | Macrophytes |
| Poz-59922 | LEM13-3A-3U | 2.34 | 14,000 | 70 | 10,290 | 10,707 | 9,850 | Macrophytes |
| Poz-59923 | LEM13-3A-3U | 2.59 | 15,630 | 130 | 12,467 | 12,997 | 11,465 | Macrophytes |
| Poz-57281 | LEM13-20m | – | 2,370 (∗∗∗) | 30 | – | – | – | DIC |
| Poz-60705 | LEM135D | – | 2,380 (∗∗∗) | 30 | – | – | – | Modern macrophytes |



**Table 2.** Sedimentological, compositional characteristics and depositional environment of LdM lacustrine facies.

**Banded diatomaceous silts (TOC <2%; BioSi 18-19 %)**

| Facies | Sedimentological properties | Composition and geochemistry | Depositional environment |
|---|---|---|---|
| **D1** | Dark to light brown, diatomaceous fine silt in cm- thick layers with gradational boundaries. | Low TOC (1.9 %), TIC (< 0.1 %) and TS (0.24 %) and moderately low BioSi (19.5 %). High silicate-related elements (K, Ti, Rb, Sr and Zr); low Br/Ti, Ca/Ti, S/Ti and Fe/Mn. | Distal, profundal environments with intermediate clastic-volcanic input (ash fall processes and clastic run-off from the watershed after volcanic events).Higher clastic input in D2. |
| **D2** | Dark grey to brown, diatom-aceous, medium to coarse silt in cm-thick layers with gradational top and bottom boundaries. | Lower TOC (1.6 %), TIC (< 0.1), TS (0.22 %) and BioSi (18.2 %) than facies D1. High silicate-related elements (K, Ti, Rb, Sr and Zr); low Br/Ti, Ca/Ti, S/Ti and Fe/Mn. | Distal, profundal environments with higher clastic - volcanic input |

**Banded, diatomaceous, organic-rich silts (TOC >3%; BioSi > 20%)**

| Facies | Sedimentological properties | Composition and geochemistry | Depositional environment |
|---|---|---|---|
| **D3** | Banded, dm-thick layers of dark brown diatomaceous, organic rich medium to coarse silts with variable macrophyte content. | High TOC (3.8 %) and BioSi (21.7 %), moderate TS (0.6 %) and low TIC (<0.1 %). Relatively high Si, Fe and Br and low Ti, Rb, Sr, Zr, Ti, Br/Ti, Ca/Ti, Fe/Mn. | Littoral to relatively deep environments dominated by organic deposition with low clastic input. Sequences (D3/D5,D4), (D5/D3/D4) and alternation D3/D4 represent transition from littoral, macrophyte-dominated (D3 and D5) to deeper environments (D4). |
| **D4** | Massive to banded, < dm-thick layers of lighter brown diatom and organic rich fine silt. | High TOC (3.2%) and BioSi (21.4 %), moderate TS (0.7 %) and low TIC (< 0.1 %). Relatively high Si, Fe and Br and low Ti, Rb, Sr, Zr. | |
| **D5** | Massive to banded, mm- to cm - thick layers of brown, macrophyte-rich coarse silt. | TS (1.4 %) and presence of calcite (TIC = 0.1%). Higher TOC/TN. Moderate but fluctuating silicate-related elements (K, Ti, Rb, Sr and Zr), high Br/Ti, Ca/Ti, and S/Ti. High Mn values, and low Fe/Mn ratios. | |

**Laminated diatomaceous organic-rich silts(TOC 2-3.5%; BioSi ca. 20 %)**

| Facies | Sedimentological properties | Composition and geochemistry | Depositional environment |
|---|---|---|---|
| **D6** | Laminated, dark brown, organic-rich silt composed of three main laminae type: a) greenish grey, diatomaceous medium to fine silt; b) dark brown, organic-rich fine silt with diffuse upper and lower boundaries and c) brown, macrophyte–rich, coarse silt, with sharp, erosional lower boundaries. Facies D6b contains more frequent endogenic calcite crystals. | High TOC (3.1%), BioSi (19.7%) and TS (1.4%, peaks up to 6% in D6c) and common presence of calcite (TIC >= 0.1). Moderate but fluctuating silicate-related elements (K, Ti, Rb, Sr and Zr); high Br/Ti, Ca/Ti, and S/Ti. High Mn values and low Fe/Mn ratios. | Distal, frequently oxic environments dominated by organic deposition and higher clastic input. Some periods with endogenic calcite formation |



**Table 3.** Depositional and compositional characteristics of LdM mass wasting facies.

| Facies | Sedimentological properties | Composition and geochemistry | Depositional environment |
|---|---|---|---|
| LT1 | Up to 1.1 m thick layer composed of massive dark grey coarse silt with dispersed white mm-long pumice clasts and macrophyte remains. At the base, some faintly banded intervals (> dm-thick) with higher organic content. | Moderate TOC (2.2 %) and low TIC (0.3 %), TS (0.4 %) and BioSi (17.7 %). High silicate-related elements (Si, K, Ti, Rb, Sr and Zr); low Br/Ti, Ca/Ti, S/Ti and Fe/Mn. | Lacustrine turbidite deposited in distal areas. Abundance of macrophyte suggests littoral areas as sediment source. |
| LT2 | More than 60 cm thick layer composed of massive, black, medium silt with lower macrophyte content than LT1. Magnetic susceptibility is higher at the base but grading is not evident. | Lower TIC (0.1 %) and BioSi (15.1 %) and higher TOC (X %), TS (1.0 %) than LT1. Large C/N range. High silicate-related elements (Si, K, Ca,Ti, Rb, Sr and Zr); low Br/Ti, Ca/Ti, and Fe/Mn and moderate S/Ti. | Lacustrine turbidite deposited in distal areas. Absence of macrophyte suggests sublittoral, deeper areas as main sediment provenance. |

**Table 4.** Depositional and compositional characteristics of LdM volcanic facies.

| Facies | Sedimentological properties | Composition and geochemistry | Depositional environment |
|---|---|---|---|
| T | Tephra: Banded to laminated, dark grey to black, cm thick tephra layers mostly composed of volcanic glass, quartz and plagioclase and with sharp basal and top boundaries and fining upward textures. | Low TOC (1.9 %), TIC (< 0.1%), TS (0.2%) and BioSi (15.0%) Two main groups from XRFdata: 1) tephras (T1, T2, T4, T5, T11 and T14) with higher Sr, Ti, Fe and lower Si and Rb and 2) tephras (T3, T6, T7, T8, T9, T10, T12, T13, T15, T16, T17 and T18) with higher Si and Rb and lower Fe and Sr. | Fine ash fall deposits |
| L | Lapillis: Massive to banded coarse grey lapilli in layers up to 20 cm. Younger L1 and L2 with MS > 200 SI and L3, L4 and L5 with MS < 100 SI. | | Coarse fall volcanic deposits |
| CT | Cryptotephras: Massive to faintly banded, dark grey to black, cm-thick coarse silt composed of volcanic fraction (quartz, plagioclase, glass), organic matter and diatoms. Upper and lower boundaries are diffuse. | | Reworked volcanic material from the watershed transported into the distal areas of the lake by fluvial and run-off processes. They only occur in unit 2. |





**Table 5.** Age estimated for LdM tephra.

| ID | Depth(m) | $^{14}$C cal a BP | lower a | upper a | Thickness mean (cm) |
|---|---|---|---|---|---|
| **T1** | 0.13 | 18 | 22 | 13 | 1.5 |
| **T2** | 0.43 | 650 | 757 | 571 | 3.0 |
| **T3** | 0.51 | ~680 | ~800 | ~590 | 4.0 |
| **T4** | 0.59 | ~680 | ~800 | ~590 | 5.5 |
| **T5** | 1.13 | 2080 | 2623 | 1585 | 1.0 |
| **L1** | 1.22 | 2285 | 2840 | 1760 | 7.0 |
| **T6** | 1.36 | 2490 | 3040 | 1959 | 1.0 |
| **L2** | 1.77 | 3659 | 4080 | 3185 | 4.0 |
| **T7** | 1.97 | ~4050 | ~4400 | ~3760 | 4.0 |
| **T8** | 2.74 | ~4050 | ~4400 | ~3760 | 2.0 |
| **L3** | 2.77 | ~4050 | ~4400 | ~3760 | 16.0 |
| **T9** | 2.96 | ~4050 | ~4400 | ~3760 | 7.0 |
| **T10** | 3.10 | 4570 | 5206 | 4120 | 3.0 |
| **T11** | 3.44 | 7178 | 7650 | 6410 | 2.0 |
| **T12** | 3.60 | 8300 | 8735 | 7890 | 1.0 |
| **T13** | 3.63 | 8460 | 8735 | 7890 | 1.0 |
| **L4** | 3.65 | 8685 | 9063 | 8440 | 1.0 |
| **T14** | 3.98 | 10780 | 11314 | 10440 | 1.0 |
| **T15** | 4.03 | 11140 | 11844 | 10625 | 1.0 |
| **T16** | 4.06 | 11325 | 12050 | 10760 | 1.0 |
| **T17** | 4.14 | 11991 | 12689 | 11240 | 1.0 |
| **L5** | 4.17 | 12185 | 12800 | 11430 | 10.0 |
| **T18a** | 4.34 | 12513 | 13015 | 11813 | 1.0 |
| **T18b** | 4.36 | 12835 | 13222 | 12175 | 3.0 |