# Peer review of "Volcanism and climate change as drivers in Holocene depositional dynamic of Laguna del Maule (Andes of central Chile - 36°S)"

_Climate of the Past, 2019_

## Referee Comment (RC1) · Anonymous Referee #1 · 16 Feb 2020

General Comments: Frugone-Álvarez et al. provide a highly detailed, multi-proxy study of a volcanic lake in the central Andes, with emphasis on geochemical analysis of downcore lake sediment. The clear organization of scientific methods and assumptions as well as the use and integration of diverse disciplines (e.g., stratigraphy, seismic surveys, geochemistry, statistics, tephra, volcanism, climate) to reconstruct a paleoclimate story make this a valuable contribution to Climate of the Past. The well-characterized framework that this paper provides for the complex LdM site sets the stage, for what I imagine, is a lot more interesting work planned by this group. I support acceptance of the manuscript upon some minor revisions outlined below.

[Figure]

Specific Comments: I noticed in the abstract, and at times throughout the text and tables, the nomenclature for describing ages varied. I'd suggest choosing one and sticking with it, such as cal a BP (and cal ka BP), which indicates that the ages discussed are calibrated and provides an appropriate reference point (i.e. BP).

Early, Middle and Late Holocene should all be capitalized as they have recently been formalized subdivisions for the Holocene (Walker et al., 2019).

How certain are the authors that the ash layer suggested to be the Quizapú tephra is indeed so? Are there other possible eruptions that could be correlated with this layer as well, especially since major (or trace) elemental analysis was not performed? I know from personal experience that assuming a tephra as a key marker without geochemical data to support it can sometimes be incorrect, and instead, turn out to be another layer all together. This seems especially important considering that one goal of this work is to lay a tephra stratigraphy framework for this region.

Can the authors place an estimated uncertainty on the reservoir effect that results in offsets of the age model? In other words, the authors assume a constant offset of 4.7 ka, and rightfully acknowledge a level of uncertainty inherent to this, however, to what extent? Could DOC in the lake ever be in equilibrium with the atmosphere during the Holocene such that there is periodically no offset, or somewhere in between? Given that the age model is used to compare against regional climate records, being clearer about this uncertainty is extremely important.

What is the threshold %TOC value for laminated organic-rich silts (L166)?

In the paleoclimate section, what ages are used to define the Medieval Climate Anomaly and the Little Ice Age? I know these events mostly as Northern Hemisphere climate anomalies. How are they expressed in the Southern Hemisphere in terms of temperature and precipitation?

L601-602: How are these rhyodacite eruptions linked to your L2 and L1 lapilli deposits?

L616-620: Why is the two-phased structure of the LIA not discussed in the main text if it is in the conclusions? Similarly, solar irradiance is brought up directly afterwards as a centennial-scale climate forcing but not discussed in the main text. Expanding on these points in the discussion would be important if they are to remain as conclusions.

Technical Corrections: L6: remove "is" and make date plural (i.e. "dates") L7: add "the" before "Early Holocene", change "were" to "was" L8: reverse order of major hydroclimate transitions (i.e. oldest to youngest) L21: add "in" before "terrestrial ecosystems" and "as well as" before "atmospheric and ocean circulation" L33: It seems like a word is missing – a hazard to regional xxx in central Chile. Add "the" before "last deglaciation". L170-172: These are not complete sentences and need some rewriting. L173: Delete "they" L175: I think it should read "Well-defined troughs in BioSi values occur at…". There are extra words here that make the meaning of the sentence unclear. L282: change "dominant" to "dominate" L282: change "occurred" to "occurs" L299: Capitalize "unit" as it is the first word of the sentence L305: remove "a" after "Establishing" L307: add "to" before "degassed" L335: add "age" after "pre-Holocene" L336: change "as" to "to" L337: change "forming" to "formed" L352: seems like a word is missing. Maybe add "although" before "…if this emplacement would have been related…" L395: add "and" before "reflect" L397-399: I think the final sentence of the paragraph should be two and read as the following "The decrease in TOC and the less abundant calcite occurrences indicate a less productive environment towards the end of this phase. Simultaneous decreases in Ephedral/Poaceae values indicate increased aridity." L519: change "shown" to "show" L542: remove "also" L546: remove the second "in" L555: add either "in/of" after "fluctuations" L564: change "concomitant to" to "concomitant with" L565: Spell out LIA here since it's the first time it is mentioned L569: This sentence is long and needs rephrasing in the beginning. L599: add "the" before "Holocene Phase"

Sources: Walker, M., Head, M.J., Lowe, J., Berkelhammer, M., Björck, S., Cheng, H., Cwynar, L.C., Fisher, D., Gkinis, V., Long, A., Newnham, R., Rasmussen, S.O., Weiss,

H., 2019. Subdividing the Holocene Series/Epoch: formalization of stages/ages and subseries/subepochs, and designation of GSSPs and auxiliary stratotypes. Journal of Quaternary Science 34, 173-186.
* * *

---

## Referee Comment (RC2) · Leonie Peti (Referee) · 18 Feb 2020

General comments

Frugone-Álvarez et al. present a thorough, multidisciplinary study on multiple sediment cores from the Laguna del Maule (LdM) lake in the Chilean Andes. The paper is rich in new multi-proxy data (Chronology, Stratigraphy, Bathymetry, Seismic, Sediment description, Tephra and Sediment micro-XRF, pollen) and extensive supplementary material building on previous investigation of a shorter record. The integration of these datasets and regional comparisons are used to derive large scale atmospheric and hydroclimatic changes in the Holocene of South America. The figures are detailed

and support the manuscript well. This paper contributes to closing the gap of our understanding of environmental and climatic changes in the Southern Hemisphere and is very suitable for Climate of the Past. I recommend publication after some minor revisions.

I have outlined some comments below. Technical corrections/suggestions on the main text and the supplementary material are attached as pdfs with comments.

Specific comments

I complement the authors on citing R packages but version numbers (on packages and R itself) should be included too.

I would really like to see the data presented in this paper shared in an online repository. "The data from proxies and facies are available from the authors upon request." is unsatisfactory and does not ensure data availability for the long-term future.

Chronology

I want to complement the authors on incorporating age uncertainty in the plots of proxy data (Figs. 7,8) and the discussion as well as considering the short-comings of the current age model.

It is unfortunate that no tephrochronology could be established, especially in such a volcanically active region. Tephrochronology has a high potential to strengthen the chronology but also to modify it with possible changes in the interpretation, especially considering the differences between the Singer et al 2008 36Cl ages and the presented 14C age model. I would suggest to highlight that the main conclusions of the paper (as the focus lies on the Mid to Late Holocene) are independent of this discrepancy to avoid discrediting the interpretation.

The age model plot in Fig. 5 needs to show the panels with iterations, accumulation rate and memory, which are included in the default output from Bacon and hold important information for the chronology development.

Why was a prior of 80 a/cm chosen as a prior for the accumulation rate? An approximate estimate of the ca. 260 cm in the LdM sequence without layers of instantaneous deposition and almost 14 ka would suggest something closer to 50 a/cm. How does the posterior distribution look (see panels in age model plot by Bacon, comment above)?

And why was a segment length of 4 cm ("thick") chosen? I understand this value is rather arbitrary but can have quite some influence on the resulting chronology. This should be discussed/acknowledged, or at least the information of those extra panels provided by Bacon should be included in Fig. 5 to be able to judge the performance of the age model.

Micro-XRF data

It is not always clear if log ratio transformed micro XRF data are used for the subsequent statistical analyses, simple ratios or raw data (see also comment in the pdf regarding ln(x) or ln(x/y) or centralised log ratio). Please double check as that may impact the results/interpretation.

Line 116-117. How was it decided to use a cut off value of 1000cps for elements to be excluded from the dataset? I imagine this has a significant influence on the interpretations, especially since some interesting elements are excluded this way. In this context, I did not understand clearly whether the volcanic facies (tephra and lapilli) and LT layers are included in the calculation of the mean. If yes, this surely favours some elements in a potentially dubious way.

Line 250 Clastic-related elements in the first eigenvector are explained mostly by silicates from the volcanic watershed. If I understand correctly, the volcanic facies were excluded, so this refers to reworked volcanic material in the other facies? But why is Si not dominant (according to the listed elements) if the detrital signal of the first eigenvector (which I agree with) is to be explained by silicates?

I am curious about the calibration between ICP-OES and micro-XRF samples. How

did the authors ensure that the correct points were compared with each other? Are the discrete samples scanned or how does one know that a discrete sample (of which thickness? Same as the micro-XRF resolution?) matches exactly with a specific scanning step? However, this is not very important (in the context of the paper) as I do not see where the calibrated fully quantitative data are used instead of just the semi-quantitative XRF core scanning data.

Volcanism

Line 541 Volcanic/seismic activity are used interchangeably. Is there any chance the authors could discriminate between the triggers?

Does the inferred change in magma composition in the Late Holocene have any impact on the depositional dynamic of LdM, the climate or societies (given the topic of the special issue this is included in)?

Technical corrections

The supplementary material has two figures named "S5" resulting in wrong references and wrong numbers for figures S6-S13. I may not have marked all occurrences of this, please double check.

Further technical corrections are marked in the attached pdf of the manuscript and supplementary material.

Please also note the supplement to this comment:
https://www.clim-past-discuss.net/cp-2019-147/cp-2019-147-RC2-supplement.zip
* * *

---

## Referee Comment (RC3) · Anonymous Referee #3 · 21 Feb 2020

Frugone-Álvarez et al. characterized and discussed a very complex stratigraphy from Laguna del Maule in central Chile. Their reconstruction is supported by a wide range of sedimentological, geochemical and biological proxies. The integration of all these datasets is used to reconstruct the local volcanic history, as well as regional paleoclimate trends and potential mechanisms. This is a very comprehensive manuscript, with regional and hemispheric implications. Unfortunately, the chronology shows significant uncertainties, although the authors correctly considered them for their paleoenvironmental inferences. I think this study represent a genuine contribution to a better understanding of past volcanic and climate changes in South America, and consequently I recommend its publication in Climate of the Past after some modifications.

[Figure]

I can recognize only one major issue in this study, which is related to the interpretation of the pollen data. A significant number of minor corrections are outlined at the end of this review.

Interpretation of the pollen data In Line 383 the authors mention that the pollen data reveals "sparse vegetation and relatively high Ephedra/Poaceae ratio would suggest relatively humid conditions facilitating an upward shift of lower vegetation belts." How can humid conditions facilitate an upward expansion of lowland vegetation? In most mountain regions humid conditions tend to promote downslope invasions of high-altitude taxa. This should be the case in the Laguna del Maule area, as rainfall increases with elevation (Supplementary Figure S4). In my opinion the authors should reconsider their vegetation-climate interpretations or, alternatively, provide supporting information.

In addition, is hard to understand how an upward expansion of lowland vegetation can be expressed by a rise in the Ephedra/Poaceae index of Figure 7. To my (rather limited) understanding of the flora of Chile, several species of the Poaceae family are commonly found in the high Andes, with their altitudinal distribution being, on average, higher than Ephedra. Can the authors state which are the relative climate affinities of Poaceae and Ephedra? I think this would clarify the interpretation of the index. There might be also a methodological problem in the actual index calculation. The pollen ratio in Figure 7 was calculated from the formula (a-b)/(a+b); where "a" corresponds to Ephedra and "b" corresponds to Poaceae. Yet, in Figure S12 Poaceae shows higher abundance than Ephedra in almost all samples (b>a). If so, shouldn't the ratio be dominated by negative values? This issue makes the understanding of this index a bit confusing. For instance, the high values seen at the beginning of Phase 3 and during Phase 6 (Figure 7) are hard to reconcile considering that these phases are actually associated with rises in Poaceae and a drop in Ephedra (supplementary Figure S12). It would be great if the authors address this issue and ensure that the index is well calculated.

Finally, although I am not sure how was the index calculated; from my understanding of the regional vegetation and the pollen data of Figure S12, the index in Figure 7 could

be directly proportional to regional humidity. In this case, high index values during pre-Holocene and the late Holocene time would indicate relative high precipitation, whilst low values during the early to mid-Holocene would reveal a drop in regional precipitation.

Minor corrections Line 6. "We produce an age model based..." Line 7. "According to this age mode, early Holocene..." An adjective for early Holocene is missing in this sentence. Line 12. "During the late Holocene, the tephra layers show..." Line 21. Consider change sentence to "...have documented major changes in the productivity of terrestrial ecosystems, atmospheric and oceanic circulation..." Line 22. "western slopes" Line 32. "..., does show that this is a regional hazard to central Chile." Line 49. "...during known rapid climate changes...". There is no need to create an acronym (RCC) if it is not going to be used again. Line 51. "70°30'W" Line 68. "CO2" Line 130. Please provide the country of the Keck Radicarbon Facililty. Line 135. It seems that radiocarbon ages were not calibrated and simply reported as conventional 14C years. This might be problematic and inconsistent with Figures 5 and 7, which have their temporal axes in the calendar age scale. Please provide an explanation to this issue. Line 139. There is no explanation of how the Quizapú ash layer was identified in the methods section. There is a mention later (Line 310), but I my opinion it should be included here. Lines 169-171. There is something missing in the sentence starting with "The finer grain size of ...". Please revise. Line 175. "Biogenic silica concentrations range from 5 to 26%...." Line 178. "Well-define peaks (throughs)...". Not clear. Line 284. "ratios" Line 284. Please provide a climate interpretation for "an upward shift to lower vegetation belts". Line 298. "the promulgations of the forest law in 1931 that had a large impact in deforestation..." How can a deforestation process be associated with a sharp increase in a tree (Pinus)? Line 300. "Unit 4" Line 307. "likely due to..." Line 338. "36Cl". Line 352. Triggering process (3) does not follow the same grammatical structure tan processes (1) and (2). Line 377. Unlike all the other variables, $\delta$13C and $\delta$15N are not shown in Figure 7. Perhaps it will be useful to include them in order to facilitate comparisons. Line 382. "(Meyers and Teranes, 2002)" Line 395. "...settings),

reflecting. . ." Line 417. ". . .beginning of this phase followed by a decreasing. . .." Line 446. ". . .exhibits centennial-scale oscillations. . ." Line 473. "carbonate-producing" Line 494. I which way a strengthening of ENSO would lead to a southward shift of the ITCZ? Line 494. Define "ITCZ" Line 509. "...a progressive increase up to. . .". A progressive increase of what? Line 519. ". . .during the early-to-mid Holocene, with clear. . ." Line 530. "mid-latitude" Line 531. "Variation in the strength of the. . ." Line 542. ". . .favors a climate influence. . ." Line 555. "Both greater fluctuations in water levels. . ." Line 565. Define "LIA", both acronym and chronozone. Line 566. Provide chronozone for Medieval Climate Anomaly.

---

## Author Comment (AC1) · 15 Apr 2020

Response to referee #1

We appreciate the exhaustive work carried out by anonymous referee #1, which have greatly helped to improve manuscript clarity. We believe we have addressed all comments and concerns of reviewer #1 and are mostly in agreement on most issues. Below, we have addressed all the reviewer's comments and explained how we have changed the manuscript accordingly.

General Comments referee #1: Frugone-Álvarez et al. provide a highly detailed, multiproxy study of a volcanic lake in the central Andes, with emphasis on geochemical analysis of downcore lake sediment. The clear organization of scientific methods and assumptions as well as the use and integration of diverse disciplines (e.g., stratigraphy, seismic sur- veys, geochemistry, statistics, tephra, volcanism, climate) to reconstruct a paleoclimate story make this a valuable contribution to Climate of the Past. The well-characterized framework that this paper provides for the complex LdM site sets the stage, for what I imagine, is a lot more interesting work planned by this group. I support acceptance of the manuscript upon some minor revisions outlined below.

R: Thank you very much for your time invested to review our manuscript. The LdM sequence is a complex record, so we really appreciate your help!

1. Specific Comments referee #1:

Specific Comments 1.1 I noticed in the abstract, and at times throughout the text and tables, the nomenclature for describing ages varied. I'd suggest choosing one and sticking with it, such as cal a BP (and cal ka BP), which indicates that the ages discussed are calibrated and provides an appropriate reference point (i.e. BP).

R: We agree and have changed all ages in the ms. to "cal ka BP."

Specific Comments 1.2 Early, Middle and Late Holocene should all be capitalized as they have recently been formalized subdivisions for the Holocene (Walker et al., 2019).

R: These are all now corrected to reflect this new nomenclature.

Specific Comments 1.3 How certain are the authors that the ash layer suggested to be the Quizapú tephra is indeed so? Are there other possible eruptions that could be correlated with this layer as well, especially since major (or trace) elemental analysis was not performed? I know from personal experience that assuming a tephra as a key marker without geochemical data to support it can sometimes be incorrect, and instead, turn out to be another layer all together. This seems especially important considering that one goal of this work is to lay a tephra stratigraphy framework for this

region.

R: We agree, we do not have evidence geochemistry that it is really the eruption of Quizapú. But, according to historical data and our superficial age model only two eruptions have occurred with a significant explosivity index near LdM during the last 200 years: the eruptions of Quizapú volcano of 1846-47 and of 1932 (Figure 1). The Quizapú eruptions of 1846-47 and 1932 were of nearly identical magma, but the first eruption was effusive and the second plinian with a VEI index = +5 (Fontijn et al., 2014; Hildreth and Drake, 1992). The upper black and grey tephra (Facies T1, $\sim$ 2 cm of thickness in the LEM11-1A core) has been identified throughout the lake basin, with a change in thickness from about 2 cm thick along the northern areas of the basin and 1 cm along the southern areas of the LdM (see Figure S13). Fortunately, the stratigraphic correlation of all short cores was easily performed by comparing TOC profiles and the key ash layer located at a similar depth between all cores, also is consistent with what is described by Hildreth and Drake, (1992) and Sernageomin (Figure 1). Although we only have FESEM-EDX (Figure 2), XRF and DRX data. Sedimentological and compositional (microscope smear slides) description of the Facies T1, the age model based on 137Cs dating supports that this layer is most likely the more recent Quizapú plinian eruption described by the Servicio Nacional de Geología y Minería (Sernageomin) (Figure 1). Therefore, we conclude that T1 has a high probability of being the tephra deposition from the great eruption of 1932.

Fontijn, K., Lachowycz, S. M., Rawson, H., Pyle, D. M., Mather, T. A., Naranjo, J. A., and Moreno-Roa, H. (2014). Late Quaternary tephrostratigraphy of southern Chile and Argentina. Quaternary Science Reviews, 89:70–84. 39 Hildreth, W. and Drake, R. E. (1992). Volcán quizapu, chilean andes. Bulletin of Volcanology, 54(2):93–125. 00134

Specific Comments 1.4 Can the authors place an estimated uncertainty on the reservoir effect that results in offsets of the age model? In other words, the authors assume a constant offset of 4.7 ka, and rightfully acknowledge a level of uncertainty inherent to this, however, to what extent? Could DOC in the lake ever be in equilibrium with the atmosphere during the Holocene such that there is periodically no offset, or somewhere in between? Given that the age model is used to compare against regional climate records, being clearer about this uncertainty is extremely important.

Reply specific comments 1.4: We greatly appreciate the reviewer's comments regarding the geochronologial aspects, as they are critical for understanding the paleoclimate implications of the record. We are very much aware of the variable level of uncertainty in the age model and we have been careful to include this uncertainty in the interpretation of the record and the comparison with other time series. As stated in the paper, the uncertainty of our age estimates was established by the dating errors in the samples from short cores and recent organic matter samples (ca. 60 years, see Table S4). This uncertainty is also clearly present in our figures (as additional shading in our proxy records). Of course we also have to consider the possible variable carbon dynamics in the lake during the Holocene resulting in a variable reservoir effect through time. The epilimnia of most lakes are well mixed tend to be 14C equilibrates between the lake and the atmosphere but in volcanic lakes it does not have to be this way. During the Holocene the DOC could be out of equilibrium at different times, it is true. The correlation with the tephras will be a step forward to understand these processes. Although we are confident that the reservoir effect likely stayed within a similar range during the Late Holocene, as the lake basin and depositional processes did not greatly changed, we could not find enough terrestrial material to date with radiocarbon in the Mid and Early Holocene – or apply other radiometric techniques – so we cannot rule out this possibility. But as a first approach we have considered that the reservoir effect remained within a similar range, as in other volcanic areas with large reservoir effects (Miscanti, Chungará). A detailed and comprehensive tephra chronology will help to produce a more robust chronology in the future.

Specific Comments 1.5 What is the threshold %TOC value for laminated organic-rich silts (L166)?

R: The threshold %TOC values for D6 are $\sim$ 0.5% to 5.5% (see Supplementary Figure

S6). Laminated facies D6 have the highest TOC and TS values, up to 5.5 % and 7.0 %, respectively. Endogenic calcite occurrences are common in D6. Subfacies are identified based on organic content (higher in D6b) and type (more algal in D6a versus more macrophyte in D6c), the nature of the lamination (better defined in D6a and D6c), and the presence of carbonate (more common in D6b).

Specific Comments 1.6 In the paleoclimate section, what ages are used to define the Medieval Climate Anomaly and the Little Ice Age? I know these events mostly as Northern Hemisphere climate anomalies. How are they expressed in the Southern Hemisphere in terms of temperature and precipitation?

R: The MCA was defined using records from California and Argentina (Stine et al., 1991). Less than a handful of records have been used to describe anomalies during the MCA and LIA from central Chile (e.g. von Gunten L. et al., 2009). They provide quantitative evidence for the presence of a MCA as warm summers between AD 1150 and 1350, and a cool period corresponding to the "Little Ice Age" starting with a sharp drop between AD 1350 and AD 1400 and ending around 1850 CE. There are even fewer precipitation reconstructions, mostly based on tree ring records. In our LdM record, a phase of increased productivity (warmer?) occurred prior to 1400 CE and it would have been during the MCA. Pollen data for the last 700 years in LdM show the highest percentages of Poaceae and High Andean Steppe taxa suggestive of a displacement toward lower elevations of the high-altitude vegetation belts from 1570–1920 AD. Although these results suggest a shift toward more wetter/colder conditions during la LIA, we remain cautious because of the low resolution of this sampling interval and the lack of local pollen rain surveys.

Specific Comments 1.7 L601-602: How are these rhyodacite eruptions linked to your L2 and L1 lapilli deposits?

R: Unfortunately, we are not able to give a precise answer as we do not have detailed geochemical analyses of the tephra layers. A collaboration in this regard is in progress.

Specific Comments 1.8 L616-620: Why is the two-phased structure of the LIA not discussed in the main text if it is in the conclusions? Similarly, solar irradiance is brought up directly afterwards as a centennial-scale climate forcing but not discussed in the main text. Expanding on these points in the discussion would be important if they are to remain as conclusions.

R: We have included the discussion of the two-phases structure of the LIA and the relationship with solar irradiance in the main text.

2. Technical Corrections

2.1 L6: remove "is" and make date plural (i.e. "dates"): Done 2.2 L7: add "the" before "Early Holocene", change "were" to "was" Done 2.3 L8: reverse order of major hydro-climate transitions (i.e. oldest to youngest) Done 2.4 L21: add "in" before "terrestrial ecosystems" and "as well as" before "atmospheric and ocean circulation" Done 2.5 L33: It seems like a word is missing – a hazard to regional xxx in central Chile. Add "the" before "last deglaciation". Done. We have changed the paragraph to: "..., does show that this is a natural hazard for the society and the economy of central Chile."

2.6 L170-172: These are not complete sentences and need some rewriting.

"Table 2 summarizes the main characteristics of the LdM lacustrine facies. The finer grain size of facies D1 and D2 and the absence of littoral components (i.e macrophyte remains), with variable clastic input, particularly higher during deposition of D2. Coarser grain size and the abundance of macrophytes remains setting for facies D3 compared to D1 and D2. Facies D3, D4 and D5 are organized in dm-thick sequences and they are macrophyte-dominated (D5, D3) to diatom-dominated environments (D4) (Fig. S5)." We have changed the sentences to: "Table 2 summarizes ... The finer grain size of facies D1 and D2 and the absence of littoral components (i.e., macrophyte remains) indicate deposition in relatively deeper water. Coarser grain size and the abundance of macrophyte remains suggest a more littoral depositional setting for facies D3 compared to D1 and D2. Facies D3, D4 and D5 are organized in dm-thick

sequences and range from macrophyte-dominated (D5, D3) to diatom-dominated environments (D4) (Fig. S5)."

2.7 L173: Delete "they" Done

2.8 L175: I think it should read "Well-defined troughs in BioSi values occur at. . .". There are extra words here that make the meaning of the sentence unclear. Done We have changed the paragraph to: Well-defined troughs in BioSi values occur at the base (432-433 cm), middle (364-366 cm), and top (313-316 cm) of unit 5, and are often associated with volcanic facies (Figure 4).

2.9 L282: change "dominant" to "dominate" Done We have changed the paragraph to: Poaceae dominate throughout, especially in unit 3 and Amaranthaceae dominate after unit 3. Ephedra is the main component principally in the part upper of unit 5. 2.10 L282: change "occurred" to "occurs" Done 2.11 L299: Capitalize "unit" as it is the first word of the sentence Done 2.12 L305: remove "a" after "Establishing" Done 2.13 L307: add "to" before "degassed" Done 2.14 L335: add "age" after "pre-Holocene" Done 2.15 L336: change "as" to "to" Done 2.16 L337: change "forming" to "formed" Done 2.17 L352: seems like a word is missing. Maybe add "although" before ". . .if this emplacement would have been related. . ." Done 2.18 L395: add "and" before "reflect" Done 2.19 L397-399: I think the final sentence of the paragraph should be two and read as the following "The decrease in TOC and the less abundant calcite occurrences indicate a less productive environment towards the end of this phase. Simultaneous decreases in Ephedral/Poaceae values indicate increased aridity." Done 2.20 L519: change "shown" to "show" Done 2.21 L542: remove "also" Done 2.22 L546: remove the second "in" Done L555: add either "in/of" after "fluctuations" Done 2.23 L564: change "concomitant to" to "concomitant with" Done 2.24 L565: Spell out LIA here since it's the first time it is mentioned Done 2.25 L569: This sentence is long and needs rephrasing in the beginning. Done We have changed the paragraph to: The LdM sequence offers a continuous record of volcanic activity in the basin (Figure 5d) which can be compared to that previously described by Andersen et al. (2017) (Figure

5e). Nevertheless, the assignment of these events found in the lake sequence (Figure 3) to dated volcanic eruptions (Singer, 2014; Singer et al., 2018) is not straightforward and will require considerably more geochemical fingerprinting of individual tephras.

2.26 L599: add "the" before "Holocene Phase" Done 2.27 Sources: Walker, M., Head, M.J., Lowe, J., Berkelhammer, M., Björck, S., Cheng, H., Cwynar, L.C., Fisher, D., Gkinis, V., Long, A., Newnham, R., Rasmussen, S.O., Weiss, H., 2019. Subdividing the Holocene Series/Epoch: formalization of stages/ages and subseries/subepochs, and designation of GSSPs and auxiliary stratotypes. Journal of Quaternary Science 34, 173-186.

———————————————

[Figure]

SERNAGEOMIN
2012

Álvaro Amigo R.

Daniel Bertin U.

Gabriel Orozco L.

Location map

Historical eruptions

ESCALA 1:100.000

Referencia geodésica
Coordenadas UTM, Datum WGS84-Z19S.

Tephra accumulation (> 1cm)

Seasonal dispersion most likely according to the height of the ash column.

| Explosivity Index (VEI) | 3 | 3-4 | 4-5 |
|---|---|---|---|
| Column height (max) | 12 km | 15 km | 25 km |

**Fig. 1.** Location map and regional isopachs for compared to a VEI index = 3; 3-4; 4-5; thickness in cm.

Quizapú tephra · 04/12/2012 10:23:01

**Sample : LEM11-1A-1G 18-19 cm**

500μm · B_LEMAL_LEM11_1A_1G_18-19cm_f3x100

Processing option : All elements analysed (Normalised)

| Spectrum | In stats. | O | Na | Mg | Al | Si | P | S | K | Ca | Ti | Mn | Fe | Total |
|---|---|---|---|---|---|---|---|---|---|---|---|---|---|---|
| 1 | Yes | 56.61 | 1.61 | 1.30 | 4.16 | 19.11 | | 4.34 | 0.90 | 6.05 | 0.20 | 0.24 | 5.49 | 100.00 |
| 2 | Yes | 57.63 | 3.52 | 1.10 | 7.63 | 22.04 | 0.16 | | 0.83 | 3.23 | 0.53 | | 3.33 | 100.00 |
| 3 | Yes | 50.63 | 1.33 | 0.57 | 6.89 | 30.57 | | | 2.85 | 0.92 | | | 6.25 | 100.00 |
| 4 | Yes | 47.89 | 2.66 | 1.02 | 7.47 | 24.34 | | | 1.78 | 4.68 | 1.09 | | 9.07 | 100.00 |
| 5 | Yes | 47.88 | 1.80 | 0.44 | 6.13 | 34.63 | | | 4.77 | 1.02 | 0.41 | | 2.91 | 100.00 |
| | | | | | | | | | | | | | | |
| Max. | | 57.63 | 3.52 | 1.30 | 7.63 | 34.63 | 0.16 | 4.34 | 4.77 | 6.05 | 1.09 | 0.24 | 9.07 | |
| Min. | | 47.88 | 1.33 | 0.44 | 4.16 | 19.11 | 0.16 | 4.34 | 0.83 | 0.92 | 0.20 | 0.24 | 2.91 | |

All results in weight%

**Sample : LEM11-1A-1G 19-20 cm**

100μm · C_LEMAL_LEM11_1A_1G_19-20cm_f2x400

| Element | Weight% | Atomic% |
|---|---|---|
| O K | 51.26 | 65.50 |
| Na K | 2.50 | 2.23 |
| Al K | 6.43 | 4.87 |
| Si K | 33.34 | 24.27 |
| Cl K | 0.18 | 0.10 |
| K K | 4.25 | 2.22 |
| Ca K | 0.40 | 0.21 |
| Ti K | 0.17 | 0.07 |
| Fe K | 1.16 | 0.42 |
| Cu K | 0.30 | 0.10 |
| | | |
| Totals | 100.00 | |

Full Scale 10728 cts Cursor: 0.000 · keV

**Fig. 2.** Field-emission scanning electron microscope (FESEM) image and energy dispersive X-Ray spectroscopy (EDX) elemental analysis of tephra T1 in LEM11-1A-1G core. General Research Support Service-SAI, Univ

---

## Author Comment (AC2) · 15 Apr 2020

Response to referee #2

I write this reply on my behalf of my coauthors. We appreciate the work carried out by Leonie Peti to review our manuscript and for his helpful comments that have greatly improved it. We believe we have addressed all reviewer comments and concerns and we agree with most of these. In this document we explained how we have changed the manuscript accordingly.

General Comments referee #2: Frugone-Álvarez et al. present a thorough, multidisci-

plinary study on multiple sediment cores from the Laguna del Maule (LdM) lake in the Chilean Andes. The paper is rich in new multi-proxy data (Chronology, Stratigraphy, Bathymetry, Seismic, Sediment description, Tephra and Sediment micro-XRF, pollen) and extensive supplementary material building on previous investigation of a shorter record. The integration of these datasets and regional comparisons are used to derive large scale atmospheric and hydroclimatic changes in the Holocene of South America. The figures are detailed support the manuscript well. This paper contributes to closing the gap of our understanding of environmental and climatic changes in the Southern Hemisphere and is very suitable for Climate of the Past. I recommend publication after some minor revisions.

I have outlined some comments below. Technical corrections/suggestions on the main text and the supplementary material are attached as pdfs with comments.

1. Specific Comments referee #2:

Specific Comments 1.1 I compliment the authors on citing R packages but version numbers (on packages and R itself) should be included too.

Reply specific comments 1.1 : Done.

Specific Comments 1.2 I would really like to see the data presented in this paper shared in an online repository. "The data from proxies and facies are available from the authors upon request." is unsatisfactory and does not ensure data availability for the long-term future.

Reply specific comments 1.2 : Done. The data will be available on CP Repository as indicated in the manuscript.

Specific Comments 1.3 Chronology. I want to compliment the authors on incorporating age uncertainty in the plots of proxy data (Figs. 7,8) and the discussion as well as considering the short-comings of the current age model.

Reply specific comments 1.3 : Thank you. We are very aware the uncertainty of the

LdM chronological model and we wanted in this manuscript both, to underline it and to show how environmental and climate reconstructions still can be made.

Specific Comments 1.4 It is unfortunate that no tephrochronology could be established, especially in such a volcanically active region. Tephrochronology has a high potential to strengthen the chronology but also to modify it with possible changes in the interpretation, especially considering the differences between the Singer et al 2008 36Cl ages and the presented 14C age model. I would suggest to highlight that the main conclusions of the paper (as the focus lies on the Mid to Late Holocene) are independent of this discrepancy to avoid discrediting the interpretation.

Reply specific comments 1.4 : We greatly appreciate the comments regarding the geochronologial aspects, as they are critical for understanding the record. In order to improve the reading and structure of the ms., in the revised version the conclusions section has been changed following the comments of the reviewer as:

"The LdM record provides a high-resolution reconstruction of depositional evolution of a volcanic lake in the South-Central Andes during the Holocene based on sedimentological, geochemical and biological indicators from a sediment core. The composite LdM sediment sequence includes distal lacustrine, volcanic and massive wasting deposits. Six lithostraphic units have been defined in the northern area of the basin and correlated with five seismic units. Lacustrine Turbidite LT2 (Unit 6) is composed of massive black silt facies whereas LT1 sediments (Unit 4) are browner, coarser material with abundant macrophyte remains and also mm-size pumice clasts. Biogeochemical differences between LT1 and LT2 could imply different depositional processes (seismic and volcanic) and/ or provenance. Although some discrepancies between our age model and the dating of volcanic episodes (Singer et al., XXXX) remain concerning the timing of some major hydrological events (blockage of the outlet by lava flows, highest lake level indicated by shorelines around the lake and drainage of the lake), the composite LdM sequence spans the Holocene, after the catastrophic drainage of the lake basin likely due to upstream erosion of the Maule River. Volcanic facies occur as lapilli

(6 layers) and ash (23 layers). Their compositional features suggest a late Holocene transition towards more silica-rich magma compositions. In spite of the chronologic uncertainties, the LdM record indicates lower lake levels during the early Holocene with millennial scale bioproductivity changes coherent with lower summer insolation and increased aridity. Higher bioproductivity occurred during the Middle Holocene (from ca. 8.0 to 6.0 cal ka BP), synchronous with the phase of aridity described for the tropical and temperate latitudes of South America. During the Middle to Late Holocene, the LdM record indicates relatively higher lake levels, consistent with increased moisture after 4.0-3.0 cal ka BP, caused by the inception of the current ENSO/PDO-like dynamics in central Chile. The Medieval Climate Anomaly is characterized by increased bioproductivity whereas the Little Ice Age shows a two-phase structure with cold/wet intervals between CE 1300–1450 and CE 1600–1850 interrupted by a warmer climate between CE 1450-1600. The LdM record also suggests that millennial-scale Holocene climate and water availability in central Chile was largely ruled by variations in the summer insolation. Complex interrelations between solar irradiance and dynamic changes in regional patterns of internal climate variability such as the ENSO/PDO-like, SWW and the SPA, however, seem to exert a major control at centennial to decadal scales."

Specific Comments 1.5 The age model plot in Fig. 5 needs to show the panels with iterations, accumulation rate and memory, which are included in the default output from Bacon and hold important information for the chronology development.

Reply specific comments 1.5 : Done

Specific Comments 1.6 Why was a prior of 80 a/cm chosen as a prior for the accumulation rate? An approximate estimate of the ca. 260 cm in the LdM sequence without layers of instantaneous deposition and almost 14 ka would suggest something closer to 50 a/cm. How does the posterior distribution look (see panels in age model plot by Bacon, comment above)?

Reply (see below)

Specific Comments 1.7 And why was a segment length of 4 cm ("thick") chosen? I understand this value is rather arbitrary but can have quite some influence on the resulting chronology. This should be discussed/acknowledged, or at least the information of those extra panels provided by Bacon should be included in Fig. 5 to be able to judge the performance of the age model.

Reply specific comments 1.6 and 1.7: We greatly appreciate the reviewer's comments regarding the geochronologial aspects, as they are critical for the climate and environmental implications of the record. All depth-age models need to at least define some parameter (a priori information) to generate it. In Bacon, the accumulation rate prior is a gamma distribution defined by two parameters: mean (named acc.mean in Bacon, and provided by the user in yr/cm), and shape (acc.shape). Hence, we have prioritized that our model crosses the last dating of the record. These parameters are: acc.mean=80; res=20; acc.shape=1.5; mem.strength=4; mem.mean=0.7 and thick=4. In any case, small changes in the parameters "acc.mean", "acc.shape" and "thick" do not generate important differences in our model (see figure below). Although all models should be tested using any sensitivity analysis to evaluate the magnitude of the effect of the different parameters on the error, and to explore potential interactions among parameters (see Valero-Garcés et al., 2019) .

Valero-Garces, B. L., González-Sampériz, P., Gil-Romera, G., Benito, B. M., Moreno, A., Oliva-Urcia, B., ... & Arnold, L. J. (2019). A multi-dating approach to age-modelling long continental records: The 135 ka El Cañizar de Villarquemado sequence (NE Spain). Quaternary Geochronology, 54, 101006.

Specific Comments 1.8 Micro-XRF data. It is not always clear if log ratio transformed micro XRF data are used for the subsequent statistical analyses, simple ratios or raw data (see also comment in the pdf regarding ln(x) or ln(x y) or centralised log ratio). Please double check as that may impact the results/interpretation.

Reply specific comments 1.8 : We appreciate your comments. We used the ln(x),

where x is a XRF raw data or ratio without transformation .

Specific Comments 1.9 Line 116-117. How was it decided to use a cut off value of 1000cps for elements to be excluded from the dataset? I imagine this has a significant influence on the interpretations, especially since some interesting elements are excluded this way. In this context, I did not understand clearly whether the volcanic facies (tephra and lapilli) and LT layers are included in the calculation of the mean. If yes, this surely favours some elements in a potentially dubious way.

Reply specific comments 1.9 : The AVAATECH X-Ray Fluorescence II core scanner has a higher detection limit with the lighter elements as Al or Si, whence we have considered data lower to 1000 (cps) as not reliable. Furthermore, the XRF scanner has other detection problem related to grain size, air and water content in the sediment matrix. Therefore, we use counts per second greater than 1000 cps to analyze the elements that are least affected by these processes. Then, we use multivariate analysis to understand the behavior of the most abundant elements in the matrix in all facies, including the volcanics. However, the statistical treatment of data considers each type of facies separately so we try to differentiate the volcanic versus sedimentary processes in them.

Specific Comments 1.10 Line 250 Clastic-related elements in the first eigenvector are explained mostly by silicates from the volcanic watershed. If I understand correctly, the volcanic facies were excluded, so this refers to reworked volcanic material in the other facies? But why is Si not dominant (according to the listed elements) if the detrital signal of the first eigenvector (which I agree with) is to be explained by silicates?

Reply specific comments 1.10 : The eigenvector associated with the higher eigenvalue is interpreted as the signal from watershed sediments that are transported into the lake and that are mostly fine volcanic material. The Si (PC1 loadings: 0.26; PC2 loadings: 0.0210; PC3 loadings: 0.4590) not important explaining the first two PCAs, so the program automatically excludes the presence of Si in the figure S7 but it appears

in table S2. Si in the sediment comes from silicates but in many facies mostly from diatoms, so the interpretation of the Si XRF data is complex. We have decided to use for BioSi and not Si XRF in the discussion as indicator of diatom productivity and not use Si XRF as indicator of silicate input due to the double source (silicates and diatoms).

Specific Comments 1.11 I am curious about the calibration between ICP-OES and micro-XRF samples. How did the authors ensure that the correct points were compared with each other? Are the discrete samples scanned or how does one know that a discrete sample (of which thickness? Same as the micro-XRF resolution?) matches exactly with a specific scanning step? However, this is not very important (in the context of the paper) as I do not see where the calibrated fully quantitative data are used instead of just the semi-quantitative XRF core scanning data.

Reply specific comments 1.11: We use the average data corresponding to the cm where e.g the organic carbon had been analyzed. We realized that this generated an added error, which is why we gave up this method to calculate concentration calibrations and we used semi-quantitative XRF data.

Specific Comments 1.12 Volcanism. Line 541 Volcanic/seismic activity are used interchangeably. Is there any chance the authors could discriminate between the triggers?

Reply specific comments 1.12: In future work we will try to address this issue. A study focused only on this point is necessary, which was beyond the objectives of this work. Microfacies and microstructures analysis could potentially enable the recognition of the triggers. It seems that local earthquakes during large eruptions could have generated the destabilization of littoral zone in the lake responsible for the turbidites but events earthquake-induced due, for example, to large mass wastings or by intraplate (intraslab or crustal) earthquakes associated to a Holocene uplift and Troncoso Fault cannot be ruled out.

Specific Comments 1.13 Does the inferred change in magma composition in the Late

Holocene have any impact on the depositional dynamic of LdM, the climate or societies (given the topic of the special issue this is included in)?

Reply specific comments 1.13: This is also beyond the scope of our study, as more detailed geochemical analyses are needed to fully understand the volcanic processes. We have added in the introduction and discussion the possible impacts of change on the depositional dynamic of LdM in the climate or societies.

—————————————————

[Figure]

[Figure]

**Fig. 1.** Evaluating the magnitude of the effect of the different parameters in age-depth model

---

## Author Comment (AC3) · 15 Apr 2020

Response to referee #3

I write this reply on my behalf of my coauthors. We appreciate the work carried out by referee #3 to review our manuscript and for his/her helpful comments that have improved our work. We believe we have addressed all comments and concerns of the reviewer and we have agreed with them. In this document we explained how we have changed the manuscript accordingly.

1. General Comments referee #3:

I can recognize only one major issue in this study, which is related to the interpretation of the pollen data.

1.1 General Comments referee #3:

Interpretation of the pollen data In Line 383 the authors mention that the pollen data reveals "sparse vegetation and relatively high Ephedra/Poaceae ratio would suggest relatively humid conditions facilitating an upward shift of lower vegetation belts." How can humid conditions facilitate an upward expansion of lowland vegetation? In most mountain regions humid conditions tend to promote downslope invasions of high-altitude taxa. This should be the case in the Laguna del Maule area, as rainfall increases with elevation (Supplementary Figure S4). In my opinion the authors should reconsider their vegetation climate interpretations or, alternatively, provide supporting information.

Reply general comments 1.1 :

The sentence was not well written, and we agree with the reviewer that it was a source of confusion. Modern pollen rain studies in northward and southward sites of LdM show a clear altitudinal relationship between Poaceae and Ephedra (Fernández Murillo et al 2019; Páez et al 1997). The first taxa are abundant in the Mediterranean Andean Shrubland dominated by Laretia acaulis and Berberis empetrifolia and it is correspondent to a vegetation belt where LdM is located; Ephedra is present in the Mediterranean Andean shrubland dominated by Chuquiraga oppositifolia and Discaria articulata, and this vegetation belt locates just below the altitude of the belt described above (Luebert & Pliscoff 2006; Figure S5). In both cases, Poaceae and Ephedra are not the main component of the vegetation, however, they are ones of few taxa in the Andean flora with anemophilous pollen dispersion syndrome (high production and dispersal of pollen). In this work, we postulate that these taxa are good indicators of each vegetation belt. So, high values of the pollen ratio, would suggest expansion of the high altitude belt and retraction of the low elevation vegetation belt and it would be associated to humid conditions. The pollen ratios have an editing problem in the manuscript

as this should be Poaceae/Ephedra instead of Ephedra/Poaceae. We have removed ratio Amaranthaceae/Poaceae and will only analyze ratio Poaceae/Ephedra. All data will be available online.

Thus, we have changed the sentence to: "Pollen samples indicate sparse vegetation and relatively high Poaceae/Ephedra ratio, suggesting expansion of high altitude belt and retraction of low elevation vegetation belt is associated to humid conditions during this period."

Páez, M.M, C. Villagrán, S. Stutz, F. Hinojosa & R. Villa. 1997. Vegetation and pollen dispersal in thesubtropical-temperate climatic transition of Chile and Argentina. Review of Palaeobotany and Palynology 96(1-2): 169-181.

Fernández Murillo, M.P., J.G. Cuevas & A. Maldonado. 2019. Análisis de la lluvia polínica actual enun gradiente altitudinal en los Andes de Chile Central (33° S). Gayana Botánica vol. 76, No. 2, 103- 119

1.2 General Comments referee #3:

In addition, is hard to understand how an upward expansion of lowland vegetation can be expressed by a rise in the Ephedra/Poaceae index of Figure 7. To my (rather limited) understanding of the flora of Chile, several species of the Poaceae family are commonly found in the high Andes, with their altitudinal distribution being, on average, higher than Ephedra. Can the authors state which are the relative climate affinities of Poaceae and Ephedra? I think this would clarify the interpretation of the index. There might be also a methodological problem in the actual index calculation. The pollen ratio in Figure 7 was calculated from the formula (a-b)/(a+b); where "a" corresponds to Ephedra and "b" corresponds to Poaceae. Yet, in Figure S12 Poaceae shows higher abundance than Ephedra in almost all samples (b>a). If so, shouldn 0 t the ratio be dominated by negative values? This issue makes the understanding of this index a bit confusing. For instance, the high values seen at the beginning of Phase 3 and during Phase 6 (Figure 7) are hard to reconcile considering that these phases are actually

associated with rises in Poaceae and a drop in Ephedra (supplementary Figure S12). It would be great if the authors address this issue and ensure that the index is well calculated.

Reply general comments 1.2 :

We greatly appreciate the reviewer's comments regarding the pollen aspects. As we have commented above, we have decided to change the pollen ratio only to Poaceae/Ephedra. This change does not generate any modification in the interpretation or the results of the manuscript. So, the Poaceae/Ephedra ratio was calculated using the formula $(a−b)/(a+b)$ where "a" is Poaceae and "b" is Ephedra. In this way, values near 1 in the pollen relationship are interpreted as an increase of humidity, while values near to -1 suggest drier conditions.

1.3 General Comments referee #3:

Finally, although I am not sure how was the index calculated; from my understanding of the regional vegetation and the pollen data of Figure S12, the index in Figure 7 could be directly proportional to regional humidity. In this case, high index values during pre-Holocene and the late Holocene time would indicate relative high precipitation, whilst low values during the early to mid-Holocene would reveal a drop in regional precipitation.

Reply general comments 1.3:

Thank you for this important question. Poaceae/Ephedra ratio between these taxa (figure 7) show high values during the first half of unit 5, between ca. 13-10 ka, suggesting relatively humid conditions. After, a trend to decrease in the pollen ratio suggest upward of low vegetation belt associated to drier conditions. After ca. 9.0 ka a slight reversion in trends suggest relatively more humid conditions until 7.5-8.0 ka. Between 6.0-4.5 ka the values are the lowest, particularly around ca. 6.5 and 5 ka, suggesting the maximum expansion of lower vegetation belts and increase of drier conditions. Ac-

cording to this indicator, the driest period occurred around 6,5 ka. The late Holocene is characterized by higher values of this pollen ratio, suggesting expansion of high altitude belt and retraction of low elevation vegetation belt, associated to more humid conditions during this period. A slight trend to increase of pollen ratio is recorded throughout the late Holocene

2. Minor corrections referee #3:

2.1 Line 6. "We produce an age model based. . ."

Reply general minor 2.1: Done

2.2 Line 7. "According to this age mode, early Holocene. . ." An adjective for early Holocene is missing in this sentence.

Reply general minor 2.2: Done

2.3 Line 12. "During the late Holocene, the tephra layers show. . ."

Reply general minor 2.3: Done

2.4 Line 21. Consider change sentence to ". . .have documented major changes in the productivity of terrestrial ecosystems, atmospheric and oceanic circulation. . ."

Reply general minor 2.4: Done

2.5 Line 22. "western slopes"

Reply general minor 2.5: Done

2.6 Line 32. ". . ., does show that this is a regional hazard to central Chile."

Reply general minor 2.6: Done

2.7 Line 49. ". . .during known rapid climate changes. . .". There is no need to create an acronym (RCC) if it is not going to be used again.

Reply general minor 2.7: Done

2.8 Line 51. "70 âŮę 30'W" Line 68. "CO2"

Reply general minor 2.8: Done

2.9 Line 130. Please provide the country of the Keck Radicarbon Facililty.

Reply general minor 2.9: Done

2.10 Line 135. It seems that radiocarbon ages were not calibrated and simply reported as conventional 14C years. This might be problematic and inconsistent with Figures 5 and 7, which have their temporal axes in the calendar age scale. Please provide an explanation to this issue.

Reply general minor 2.10: Done. We have changed the phrase to: "Radiocarbon data are reported as radiocarbon age in years before present (relative to CE 1950). Radiocarbon ages were calibrated using the Southern Hemisphere Calibration curve (SHCal13) applying the reservoir effect in the ages of aquatic organic matter inside R package Bacon v2.2 ."

2.11 Line 139. There is no explanation of how the Quizapú ash layer was identified in the methods section. There is a mention later (Line 310), but I my opinion it should be included here.

Reply general minor 2.11: According to the stratigraphic correlations between short cores and the 210Pb/137Cs age model, we assumed that T1 is the 1932 plinian eruption fro Quizapú. The Quizapú eruptions of 1846-47 and 1932 were of nearly identical magma, but the first eruption was effusive and the second plinian with a VEI index = +5 (Fontijn et al., 2014). Fortunately, the stratigraphic correlation of all short cores was easily performed by comparing TOC profiles and the key ash layer located at a similar depth in all cores, also is consistent with what is described by Hildreth and Drake, (1992) and Servicio Nacional de Geología y Minería (SERNAGEOMIN).

2.12 Lines 169-171. There is something missing in the sentence starting with "The finer grain size of . . .". Please revise.

Reply general minor 2.12: Done

2.13 Line 175. "Biogenic silica concentrations range from 5 to 26%...."

Reply general minor 2.13: Done

2.14 Line 178. "Well-define peaks (throughs). . .". Not clear.

Reply general minor 2.14: Done

2.15 Line 284. "ratios"

Reply general minor 2.15: Done

2.16 Line 284. Please provide a climate interpretation for "an upward shift to lower vegetation belts".

Reply general minor 2.16: Done

2.17 Line 298. "the promulgations of the forest law in 1931 that had a large impact in deforestation. . ." How can a deforestation process be associated with a sharp increase in a tree (Pinus)?

Reply general minor 2.17: Due to the replacement of native forests by the introduction of novel exotic forest tree of eucalyptus and pinus.

2.18 Line 300. "Unit 4"

Reply general minor 2.18: Done

2.19 Line 307. "likely due to. . ."

Reply general minor 2.19: Done

2.20 Line 338. "36Cl".

Reply general minor 2.20: Done

2.21 Line 352. Triggering process (3) does not follow the same grammatical structure

tan processes (1) and (2). Reply general minor 2.21: Done

2.22 Line 377. Unlike all the other variables, $\delta$13C and $\delta$ 15N are not shown in Figure 7. Perhaps it will be useful to include them in order to facilitate comparisons.

Reply general minor 2.22: Due to limitations of space we have included these data in the figure S12

2.23 Line 382. "(Meyers and Teranes, 2002)

Reply general minor 2.23: Done

2.24 Line 395. ". . .settings), reflecting. . ."

Reply general minor 2.24: Done

2.25 Line 417. ". . .beginning of this phase followed by a decreasing. . .."

Reply general minor 2.25: Done

2.26 Line 446. ". . .exhibits centennial-scale oscillations. . ."

Reply general minor 2.26: Done

2.27 Line 473. "carbonate-producing"

Reply general minor 2.27: Done

2.28 Line 494. I which way a strengthening of ENSO would lead to a southward shift of the ITCZ? Reply general minor 2.28: One possibility is through the mechanism explained by Schneider et al., 2014. due to a change in the atmospheric energy balance. Averaged over a span of longitudes wide enough that one can focus on meridional fluxes, the ITCZ can be expected to lie near the "energy flux equator". Ocean energy uptake in boreal winter is approximately 15Wm2 smaller during El Niño than during La Niña, with the largest changes in the eastern Pacific. This reduction in ocean energy uptake increases, which in the zonal mean in boreal winter more than doubles from typical La Niña to El Niño. It implies that the displacement of the energy flux equator and of the ITCZ away from the Equator is reduced by more than a factor of two - sufficient to account for the observed shift in ITCZ position. The effect of changes in the atmospheric moist static energy flux equator F0 is smaller (about a quarter of the total shift). Thus, ENSO variations of the ITCZ position appear to be primarily driven by tropical changes in the atmospheric energy input. (Schneider et al., 2014)

Schneider, T., Bischoff, T., and Haug, G. H. (2014). Migrations and dynamics of the intertropical convergence zone. Nature, 513(7516), 45–53. doi:10.1038/nature13636

2.29 Line 494. Define "ITCZ"

Reply general minor 2.29: Done

2.30 Line 509. "...a progressive increase up to. . .". A progressive increase of what?

Reply general minor 2.30: "...a progressive increase in effective moisture occurred around 5.7 ka with the development of a fresher water lake..."

2.31 Line 519. ". . .during the early-to-mid Holocene, with clear. . ."

Reply general minor 2.31: Done

2.32 Line 530. "mid-latitude"

Reply general minor 2.32: Done

2.33 Line 531. "Variation in the strength of the. . ."

Reply general minor 2.33: Done

2.34 Line 542. ". . .favors a climate influence. . ."

Reply general minor 2.34: Done

2.35 Line 555. "Both greater fluctuations in water levels. . ."

Reply general minor 2.35: Done

2.36 Line 565. Define "LIA", both acronym and chronozone.

Reply general minor 2.36: Done

2.37 Line 566. Provide chronozone for Medieval Climate Anomaly.

Reply general minor 2.37: This phrase has been deleted

―――――――――――

---

## Author Response (AR1)

Pontificia Universidad Católica de Chile
Alameda 340, Santiago,
Chile, CP 6513677
(562) 2354-2610
(562) 2354-2621 FAX

Santiago, may 06, 2020

To the Editor
Climate of the Past

Enclosed pleased find the revised version of our manuscript entitled "Volcanism and climate change as drivers in Holocene depositional dynamic of Laguna del Maule (Andes of central Chile – 36° S)".

I write this letter on my behalf and that of my coauthors. First of all, we appreciate the opportunity to review our manuscript and want to thank the exhaustive work carried out by the three reviewers. Their comments helped to greatly improve the manuscript. We have addressed all comments and concerns of all reviewers and we have agreed with them in all cases. In our reply, you will find a point-by-point detailed response of the last individual questions and suggestions raised by reviewers. We are confident that we have addressed all the issues in detail and that the overall quality and novelty of the manuscript has improved and is now suitable for publication in Climate of the Past.

Sincerely,

Dr. Matías Frugone-Álvarez on behalf of all authors

Departamento de Ecología Pontificia Universidad Católica de Chile,
Instituto Pirenaico de Ecología, Procesos Geoambientales y Cambio Global and Laboratorio Internacional de Cambio Global (LINCGlobal PUC-CSIC),
Email: matutefrugone@gmail.com

Response to reviewer's comments

1) Reviewer 1

Comment "*How certain are the authors that the ash layer suggested to be the Quizapú tephra is indeed so? Are there other possible eruptions that could be correlated with this layer as well, especially since major (or trace) elemental analysis was not performed?*

*The authors did respond to the comment – and the rationale is very reasonable; however, an extra sentence in the text saying (something like) that trace/elemental analysis would be required to definitively establish the provenance… but (and then fill in the rationale that its the only reasonable conclusion, which is in the response to reviewer). ADD ONE SENTENCE TO TEXT*"

**Reply 1: First, we would like to thank our reviewer for devoting his time to study our work and preparing a high quality assessment of its merit. We have modified the paragraph to:**

...**"Although the top ash layer has not been geochemically fingerprinted, its estimated age according to $^{210}$Pb techniques coincides with the Quizapú Volcano eruption (CE 1932) and it can be used as a chrono-stratigraphic marker in the sequence (Carrevedo et al., 2015).Trace/elemental analysis would be required to definitively establish the provenance of this tephra layer the available compositional data (XRF, DRX and microscope smear slides observations) and the age mode, support our assumption that this layer is the younger Quizapú pinion eruption."**

2) Reviewers 1&2

Comments: "*Comments wrt age model uncertainty. The authors' responses are adequate – however, simply labeling the figures more explicitly (i.e., figure 8e, state shading indicates age model uncertainty etc.) would eliminate this concern; particularly since this shading was understood to be uncertainty by reviewer 2 but not reviewer 1. ADD LABELS IN FIGURES FOR UNCERTAINTY SHADING*"

        **Reply 2: We agree with the reviewers, we have added a footnote of each figure (7 and 8) as follows: "Shading indicates age model uncertainty.***"*